

Earth System
Open Access Science Discussions
Data

# The ADRIREEF database: a comprehensive collection of natural/artificial reefs and wrecks in the Adriatic Sea

Annalisa Minelli[1], Carmen Ferrà[1*], Alessandra Spagnolo[1], Martina Scanu[1], Anna Nora Tassetti[1], Carla Rita Ferrari[2], Cristina Mazziotti[2], Silvia Pigozzi[2], Zrinka Jakl[3], Tena Šarčević[3], Miranda Šimac[3], Claudia Kruschel[4], Dubravko Pejdo[4], Enrico Barbone[5], Michele De Gioia[5], Diego Borme[6], Emiliano Gordini[6], Rocco Auriemma[6], Ivo Benzon[7], Đeni Vuković-Stanišić[7], Sandi Orlić[8], Vlado Frančić[9], Damir Zec[9], Ivana Orlić Kapović[9], Michela Soldati[10], Silvia Ulazzi[10], Gianna Fabi[1]

[1]CNR-IRBIM, Largo Fiera della Pesca 1, 60125 Ancona (Italy)
[2]ARPAE (Regional Agency for Prevention, Environment And Energy In Emilia Romagna), Via Po 5, 40139 Bologna (Italy)
[3]SUNCE (Association for Nature, Environment and Sustainable Development), Obala Hrvatskog Narodnog Preporoda 7/III, 21000, Split (Croatia)
[4]University of Zadar, Mihovila Pavlinovica 1, 23000 Zadar (Croatia)
[5]ARPA Puglia (Regional Agency for Environmental Prevention and Protection of the Puglia Region), Corso Trieste 27, 70126 Bari (Italy)
[6]National Institute of Oceanography and Applied Geophysics - OGS, Borgo Grotta Gigante 42/C, 34010 Sgonico (Italy)
[7]RERA (Public Institution RERA SD for Coordination and Development of Split Dalmatia County), Domovinskog Rata 2, 21000 Split (Croatia)
[8]Ruđer Bošković Institute, Bijenicka 54, 10000 Zagreb (Croatia)
[9]University of Rijeka, Faculty of Maritime Studies, Studentska 2, 51000 Rijeka (Croatia)
[10]Municipality of Ravenna, Piazza del Popolo 1, 48121, Ravenna (Italy)

*Correspondence to*: Carmen Ferrà (carmen.ferravega@cnr.it)

**Abstract.** The paper presents a database of information on wrecks, natural and artificial reefs located in the Adriatic Sea, collected within the framework of the Interreg Italy-Croatia project ADRIREEF - Innovative exploitation of Adriatic Reefs in order to strengthen Blue Economy. The data collection lasted more than one year and comprehended three surveys and a wide literature review. After being collected, data were harmonized and, where possible, made machine-readable. Moreover, data were widely metadated, published in a webGIS (https://adrireef.github.io/sandbox3/) and shared as Open Data in EMODnet (European Marine Observation Data network) through the SEANOE repository (Ferrà et al., 2020; http://doi.org/10.17882/74880). The database is composed of 285 three-dimensional records, each one described by 51 attributes. Parameters are clustered in four main groups: identification, reef and site description, management/exploitation information. Available literature (scientific and/or grey) was also included in the database and linked to the corresponding site.



## 1 Introduction

In the Blue Innovation concept, the attractiveness and possible use of existing marine resources which are not yet fully exploited is relevant to promote sustainable economic development (Vogt, 1998; Orams, 2002; Cater and Cater, 2007; Kiper, 2013; Sakellariadou and Kostopoulou, 2015; Nurhayati et al., 2019) and redefine marine fisheries (Pauly, 2018; Stead, 2018). In this context, the recognition of less known and appreciated natural reefs (NRs), existing artificial structures (e.g. artificial reefs, rig-to-reefs; ARs) and wrecks could also be a successful way to pursue Blue Growth as all these sites might be suitable places for developing or improving sustainable activities such as fishing (both recreational and professional), nautical tourism, diving and aquaculture (Wilhelmsson et al., 1998; Stolk et al., 2007; Uyarra et al., 2009; Needham, 2010; Edney, 2011; Spalding et al., 2017; Alempijević and Kovačić, 2019). Therefore, qualitative and quantitative information about the heritage presence and already existing human activities are fundamental to pursue an ecosystem-based sea use management according to the Marine Spatial Planning Directive (EU Directive 2014/89/UE; Douvere, 2008; Gilliland and Laffoley, 2008). As MSP is based on the three pillars for sustainable development – social, economic and environmental – its implementation will facilitate in turn the enforcement of the Marine Strategy Framework Directive (MSFD, European Commission, 2008) and the achievement of Good Environmental Status (GES).

Basing on the above considerations, this paper presents a comprehensive collection of 285 natural reefs, artificial reefs and wrecks located in the Adriatic Sea resulting from the joint effort of Italian and Croatian partners participating in the Interreg Italy-Croatia project ADRIREEF (Innovative exploitation of Adriatic Reefs in order to strengthen blue economy, https://www.italy-croatia.eu/web/adrireef) aimed at assessing the potentiality of reefs in the Adriatic Sea in order to strengthen the Blue Economy.

An analogous attempt of inventory was made on intentionally sunk shipwrecks to serve as ARs over six continents (Ilieva et al., 2019). Anyway, no wreck was signalled in the Adriatic Sea, whereas available literature (Pivetta et al., 2012) and our search highlighted the occurrence of several accidentally sunken ships which, anyway, act as artificial habitats. Similar attempts to geolocalize ARs and wrecks were made along the national coasts and offshore waters of Florida and Alabama (U.S.A.), by querying municipalities about the deployment of the reefs (U.S. Fish and Wildlife Service, Wildlife & Sport Fish Restoration Program, https://www.fws.gov/wsfrprograms/) as well as in the Gulf of Mexico (Alabama Marine Resources Division, https://www.outdooralabama.com/saltwater-fishing/artificial-reefs). The latter dataset reports all the submerged structures (including ARs, rig-to-reefs and wrecks) relying on the Gulf of Mexico area, but only name, type of reef and coordinates are publicly available. Lastly, some efforts were done concerning NRs within the Reefbase project which helped localize and concisely qualify ~10000 reefs on a global scale (Oliver et al., 2002).

Another attempt made to identify aquaculture zones in the Adriatic Sea, also through a webGis application (http://www.caps2.eu/caps2/), enhances zones of production, harvesting and farms (Tara et al., 2017), but does not include some Italian reefs dedicated to this practice which are present in our database.





The collection work presented here is thus an ambitious attempt to gather in a single dataset location, geometries, history and
detailed characteristics of natural reefs, artificial reefs and wrecks existing in the Adriatic Sea. It required a deep knowledge
on the overall status of the sites, their history, past and ongoing research and monitoring programs carried out to characterize
their ecological features as well as on their current exploitation level. In this perspective, the key exercise in Activity 1 of
Work Package 3 of the ADRIREEF project was to obtain a classification of NRs, ARs and wrecks occurring in the cooperation
area and provide a map of these sites from different perspectives.
The result is an interactive map and an Open Access detailed dataset (Ferrà et al, 2020) published on EMODnet whose contents
are available for any user and purpose.

## 2 Data mining

Starting from an existing own database of the Adriatic artificial reefs, CNR-IRBIM coordinated the data collection activity to
improve it by developing and sharing among the ADRIREEF partners three online questionnaires on NRs, ARs and wrecks
(Appendix A, B and C, respectively). An initial review of available literature and data was carried out to identify the necessary
information on NRs/ARs and wrecks to be required. Questionnaires were structured in such a way to obtain a unique database
for the two reef typologies and wrecks, including physical, ecological and economic aspects and allowing to classify elements
according to their characteristics. Given the importance of collecting answers quickly and having a structured and
homogeneous database, it became essential to use easy online tools and limit the possibility of free answers by proposing
multiple-response questions (i. e. Google Forms application; https://www.google.com/forms/about/).
Wrecks were included as a specific category and deserved a dedicated set of questions since they were defined within the
project as accidentally-sunken artificial structures which are attractive for some economic activities (i.e., diving and
recreational fishing).
The collected information was used to create the final ADRIREEF database which fed, in turn, a webGis application allowing
the visualization of reefs and wrecks on an interactive map and their selection basing on running queries.
. Projects' partners (PPs) were surveyed and answers were harmonized to obtain a consistent database. New features were
defined and used as categories for the elements' classification and/or for the webGis application. Finally, all data properly
checked and harmonized were assembled and used to populate the database.

### 2.1 Literature and available data review

The review of existing literature took into account European environmental databases (https://www.eea.europa.eu/data-and-
maps, https://ec.europa.eu/environment/nature/natura2000/data/index_en.htm), research projects carried out by CNR-IRBIM
and other entities, scientific publications and grey literature. Based on the results of the review and expert knowledge, existing
gaps in the information of already known reef sites ad wrecks were identified.





With regard to ARs, a large part of inputs came from the existing CNR-IRBIM database of artificial habitats in Italy,
established in 2009 within the Italian Artificial Habitat Group of the Italian Society of Marine Biology (Fabi et al., 2011; Fabi
et al., 2015; http://www.habitatartificiali.irbim.cnr.it) and containing more than 500 bibliographic references and information
on 80 Italian artificial habitats. Bibliographic references included scientific publications and grey literature on artificial habitats
such as harbours, breakwaters, fish aggregating devices (FADs), offshore platforms and ARs since 1967. By checking this
database, it was possible to obtain a list of 150 studies regarding the Adriatic ARs published between 1977 and 2017.
Conversely, a similar literature heritage was not present for NRs and wrecks.
**2.2 Questionnaire design**
The analysis of the collected information arose the need of improving existing data especially on NRs and wrecks.
As ARs and NRs have completely different features it was decided to develop two distinct questionnaires. A third questionnaire
was developed for wrecks as they have peculiar characteristics. All questionnaires were built in a systematic way with the aims
of (i) investigating the reefs' and wrecks' suitability for Blue Economy purposes and identifying those answers that would
help to achieve this target; (ii) facilitating experts' participation in the poll by ordering questions into a logical structure.
The identification of the person/s filling in the questionnaires was considered relevant to collect consistent information and
have a contact person in case of missing data. Moreover, numerical information (distances, measures, coordinates) were asked
in specific measurement units to add collected data directly to the database avoiding transformations.
**2.2.1 Identification of required information**
**Natural and artificial reefs**
Interrogations about reefs' characteristics that could influence their suitability for sustainable exploitation were posed:
● Which is the reef and where is it located?
As a baseline, data regarding the identification of a reef are needed, therefore name and location (in WGS84
Coordinate Reference System and Decimal Degrees) of the reef were required.
● Which are the main characteristics of the area where the reef is located?
The environmental characteristics of the area where a reef is located may influence its possible exploitation as well
as its attractiveness to perform some activities, hence the following features were considered: minimum distance to
the coast (km); typology of the surrounding seabed; the presence of meadows; important biocoenoses, alien and
protected species (in case of NRs); possible protection level applied to the area (in case of NRs).
● Which are the physical features of the reef?
The reefs' physical features themselves may also influence its potential use, especially for ARs which are handmade
constructed and designed for specific scopes. To answer this fundamental question, multiple information are needed:
the typology of the reef; reef bottom depth (m); reef edge (in meters, for NRs); spatial extension of the reef (m$^2$); the





origin of the reef (for NRs); the material used for the reef construction (for ARs); structural design of the reef (for

ARs, where it is necessary to know type and number of modules/structures put in place and their layout).

●  Is the reef already managed and/or exploited and/or could it be site for new activities?

The actual use of a reef and/or the scope for which it was built, in the case of ARs, can limit the development of

further activities, thus such information is crucial to identify possible synergies and conflicts with additional potential

users. At the same time, the original purpose of an artificial reef turns out to be a key information for better

understanding monitoring and surveillance programs, management plans and possible grants taking place in the area,

as those could also limit or benefit future uses. Therefore, the following information were requested: scope/s for

which an AR was built; if the reef is managed (for both NRs and ARs) and, if yes, who is the management entity; if

a monitoring program is already in place (for both NRs and ARs) and, if yes, its duration and the investigations carried

out; if the reef area is subjected to grant or surveillance service (only for ARs). Furthermore, questions regarding

available data (scientific publications, grey literature, monitoring data) were added to the questionnaires, as they could

help for future research purposes.

**Wrecks**

Either accidentally or purposely sunken shipwrecks are full-fledged artificial structures even though they cannot be considered
as actual artificial reefs. Therefore, they were included into a specific category sharing some information with the reefs and
integrating some extra information about the physical features of wrecks.
Shared information concern (i) wreck identification (location and name), (ii) characterization of the surrounding area (distance
from the coast, type of surrounding seabed, presence of meadows), (iii) physical features of the wreck (material, bottom depth
and wreck edge), (iv) exploitation and protection of the wreck (exploitation, protection and management of the site, if existing).
Extra information asked to the partners were: weight of the wreck (tons), total area of the footprint ($m^2$), total volume of the
shipwreck ($m^3$) and known dimensions (length, width, height in meters).

**2.3 Harmonisation and construction of the database**

Firstly, all data collected from questionnaires were screened to delete duplicates and identify incomplete entries and missing
information, thus making an evaluation of a reef for Blue Economy purposes impossible. For these missing records, a data
integration was asked to the contact person.
Data collected from questionnaires were then assembled together with those already contained in the CNR-IRBIM database
and harmonized, as some answers were not in line with the requirements. Moreover, geolocations of reefs were inspected in a
GIS environment and when those were inconsistent, clarifications were asked.
Once data control and harmonization were completed, a preliminary analysis and classification of the Adriatic reefs was
performed and query filters of the webGis application were identified. Once criteria for reef classification and filters to be
applied in the webGis application were definitely agreed with PPs, the ADRIREEF database was finalized.





The final database counted for 51 columns, 48 of them derived by the questionnaires and 3 created by the database manager
(Type of reef, Country, Region). Of these fields, 10 were used as filters in the webGis application and/or for the reefs'
classification while the remaining 41 as part of technical information sheets.

**3 Database structure and geographical coverage**

The database counts 285 three-dimensional elements (Latitude/Longitude coordinate and bottom depth), described by 51
parameters and divided into 129 natural reefs, 47 artificial reefs and 109 wrecks located in the Adriatic Sea falling into Italian,
Croatian and International waters (Table 1).
All artificial reefs and most of wrecks fell within the Italian territorial waters, while the majority (79%) of natural reefs was
located within the Croatian ones. The presence of almost all the natural reefs on the eastern side of the studied area is mainly
due to the geological morphology of the Adriatic basin (Stefanon, 1972), while the complete absence of artificial reefs on the
same side is currently due to Croatian legal constraints. It is worth noting that the number of wrecks reported in the Croatian
waters is somewhat underestimated. This fact is due to the lack of basic information about several wrecks (e.g. lack of exact
position of the shipwreck, which did not allow to place it in the map), so it was decided to keep into the database only those
with adequately detailed information. It is also worthy to note that, given the great occurrence of rocky substrates along the
Croatian coast, it was agreed within the ADRIREEF Consortium to identify homogeneous areas and map each of them as a
single natural reef (Zec et al., 2019).

|  | Croatian waters | Italian waters | International waters | Total |
|---|---|---|---|---|
| **Artificial Reefs** | - | 47 | - | 47 |
| **Natural Reefs** | 102 | 27 | - | 129 |
| **Wrecks** | 9 | 87 | 13 | 109 |
| **Total** | 111 | 161 | 13 | 285 |

**Table 1: Adriatic reefs and wrecks by typology and country**.

The geographical bounding box delimiting the studied area is individuated by the coordinates: (N, S, E, W) = (46.0546,
39.4115, 20.0239, 11.7390), expressed in decimal degrees and Coordinates Reference System WGS84. The database is
available in a unique Comma Separated Values (CSV) file.
Table 2 summarizes the structure of the ADRIREEF database specifying the parameters required for each new element: current
name, relative column name, unit of measure, origin of the data (if they come directly from questionnaires or have been created
by the database manager) and possible applicability restrictions. Table 2 also reports, for each parameter, the group it belongs
to. As mentioned in paragraph 2.2.1, Group no. 1 corresponds to reef identification and geolocation information, Group no. 2





contains parameters summarizing the characteristics of the area hosting the reef or wreck, Group no. 3 concerns aspects of the
reefs/wrecks that may also have an effect on its usage, Group no. 4 includes parameters about the present and/or possible future
reef or wreck exploitation.

| Column name | Description | Group | Unit of measure | Origin of the data | Applicability restriction |
|---|---|---|---|---|---|
| type | type of reef/wreck | 1 | | DB manager | |
| country | | 1 | | DB manager | |
| region | | 1 | | DB manager | |
| location | reference city or zone for the reef/wreck | 1 | | DB manager | |
| name | common name of the reef/wreck | 1 | | Questionnaires | |
| latitude | | 1 | [decimal degrees] | Questionnaires | |
| longitude | | 1 | [decimal degrees] | Questionnaires | |
| year_deployment | year of reef deployment/wreck sink | 1 | | Questionnaires | for artificial reefs and wrecks only |
| year_modification | year of eventual modification | 1 | | Questionnaires | for artificial reefs only |
| min_depth_m | bottom depth altitude | 2 | [m] | DB manager | |
| depth_m | depth range covered by the structure | 2 | [m] | Questionnaires | |
| reef_edge_m | height of the structure | 2 | [m] | Questionnaires | for natural reefs only |
| min_dist_km | minimum distance from the coastline | 2 | [km] | Questionnaires | |
| surrounding_seabed | surrounding seabed sedimentary composition | 2 | | Questionnaires | |
| meadows | presence of meadows | 2 | | Questionnaires | |
| reef_typology | reef typology | 3 | | Questionnaires | for natural reefs only |
| structure_wreck | type of wreck | 3 | | Questionnaires | for wrecks only |
| material | material composing the reef | 3 | | Questionnaires | for artificial reefs only |
| arrangement_modules | global arrangement of modules composing the reef | 3 | | Questionnaires | for artificial reefs only |



| | | | | | |
|---|---|---|---|---|---|
| origin_reef | origin of the reef | 3 | | Questionnaires | for natural reefs only |
| total_area_sqm | total footprint area of the reef | 3 | [sqm] | Questionnaires | for natural and artificial reefs only |
| total_volume_cubm | total volume of the reef | 3 | [cubm] | Questionnaires | for artificial reefs only |
| n_oases | number of oases, composed by structures | 3 | | Questionnaires | for artificial reefs only |
| dist_between_oases_m | linear distance between oases | 3 | [m] | Questionnaires | for artificial reefs only |
| dimens_oases_sqm | footprint area of the oases | 3 | [sqm] | Questionnaires | for artificial reefs only |
| type_structures | type of structures present in the reef, composed of modules | 3 | | Questionnaires | for artificial reefs only |
| n_structures | number of structures present in the reef | 3 | | Questionnaires | for artificial reefs only |
| dim_structures_m | relevant dimensions of the structures of the reef | 3 | [m] | Questionnaires | for artificial reefs only |
| dist_between_structures_m | linear distance between structures of the reef | 3 | [m] | Questionnaires | for artificial reefs only |
| modules_shape | shape of the modules composing the structures of the reef | 3 | | Questionnaires | for artificial reefs only |
| n_modules | number of modules composing the structures of the reef | 3 | | Questionnaires | for artificial reefs only |
| dist_between_modules_m | linear distance between modules composing the structures of the reef | 3 | [m] | Questionnaires | for artificial reefs only |
| dimension_leng_width_heigh_m | dimensions L H W of the reef/wreck | 3 | [m, m, m] | Questionnaires | for artificial reefs and wrecks only |
| weight_or_displacement_tons | weight or displacement of the wreck | 3 | [tons] | Questionnaires | for wrecks only |
| experimental_professional | describes the type of exploitation, if professional or experimental, of the reef | 4 | | Questionnaires | for artificial reefs only |
| scope | describes original conception scopes of the reef | 4 | | Questionnaires | for artificial reefs only |
| exploitation | current exploitation of the reef/wreck | 4 | | Questionnaires | |
| possible_exploitation | potential exploitation of the reef/wreck | 4 | | Questionnaires | |
| observations | additional observations | 3 | | Questionnaires | for wrecks only |
| biocoenosis | presence (and types, if available) of | 2 | | Questionnaires | for natural reefs |




| | biocoenosis | | | | only |
|---|---|---|---|---|---|
| alien_species | presence (and names, if available) of alien species | 2 | | Questionnaires | for natural reefs only |
| protected_species | presence (and names, if available) of protected species | 2 | | Questionnaires | for natural reefs only |
| protected_area | presence of a protected area where the reef is placed | 2 | | Questionnaires | for natural reefs only |
| management_prog_Y_N | existing of a management program insisting on the reef/wreck | 2 | | Questionnaires | |
| management_body | name of the management body of the reef/wreck (if applicable) | 2 | | DB manager | |
| concession_area_Y_N | presence of a concession area insisting on the reef/wreck | 2 | | Questionnaires | for artificial reefs and wrecks only |
| surveillance_service | presence (and name, if available) of a surveillance service on reef/wreck | 2 | | Questionnaires | |
| current_monitoring_program_Y_N | presence of a current monitoring program on reef/wreck | 2 | | Questionnaires | |
| monitoring_programs | present or past monitoring program insisting on the reef/wreck | 2 | | Questionnaires | |
| available_data | eventually available data related to the reef/wreck | - | | Questionnaires | |
| available_literature | available literature, scientific or grey | - | | Questionnaires | |

**Table 2: Column name, the current name of the parameter, type of parameter, unit of measure, origin of the information and eventual applicability restrictions.**

## 4 Data interrogation and visualization

With the purpose of better exploiting, representing and filtering data, a webGis was created enabling the contemporary filtering (where applicable) of more than one of the following selected attributes:

- Type of element;
- Country;
- Minimum depth of the reef/wreck;
- Distance from the coastline;
- Usage of the reef/wreck;
- Reef typology (for natural reefs);
- Reef material (for artificial reefs)



The webGis main page is reachable at the address: https://adrireef.github.io/sandbox3/ and it is composed of two user-friendly
windows (Fig. 1), one reporting all available data filters (left side) and the other one (right side) showing the map where points,
identifying elements, are divided by colour in NRs (green), ARs (blue) and Wrecks (red). Hovering on an element with the
mouse, its Name and Location appear in the left bottom corner of the map.

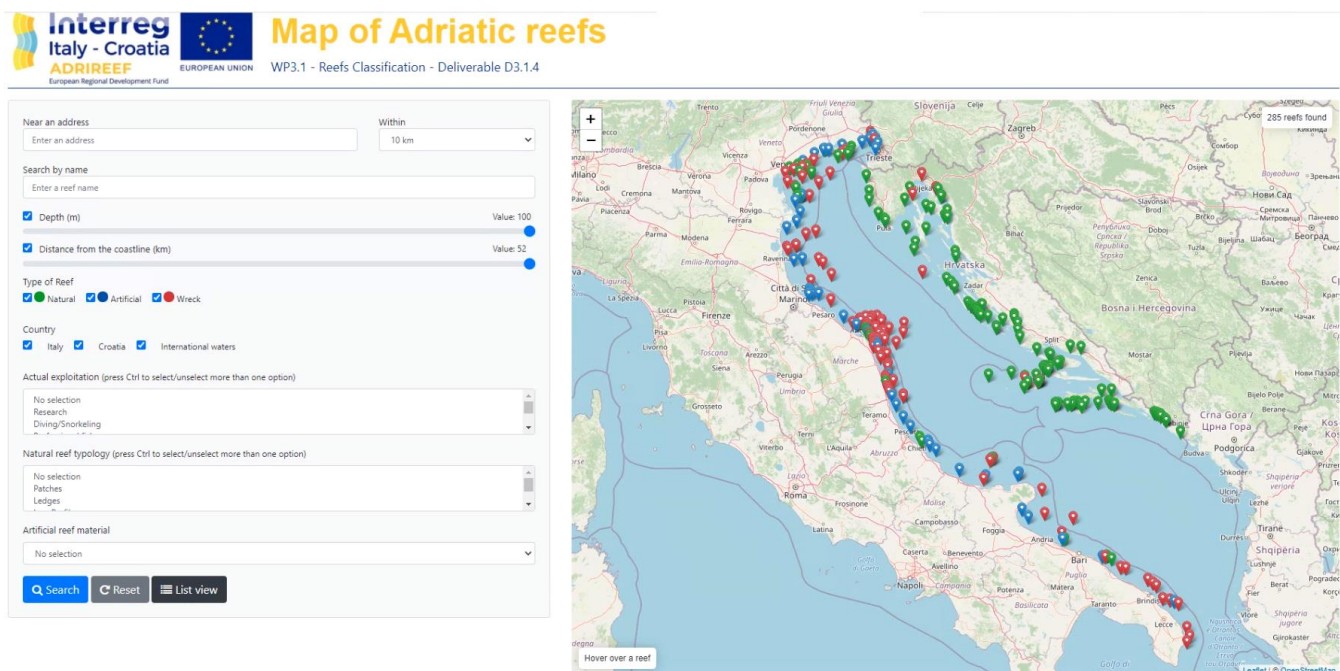


**Figure 1: The webGis interface. On the left side: the filters window. On the right side: the map window with hover function (bottom**
**left) and the total number of identified elements (top right). Basemap credits: © OpenStreetMap contributors 2020. Distributed**
**under a Creative Commons BY-SA License**
The total number of currently visualized elements is reported at the top right of the map. Moreover, when an element is clicked,
a pop-up window appears showing the associated relevant information (Fig. 2). From this pop-up, it is possible to print out
information regarding the selected element in PDF format. From the main page, it is also possible to access to the "List view"
that shows, for the visualized elements, some common information through natural, artificial reefs and wrecks (Fig. 3). The
number of visualized elements, in the top right corner of the map view, is updated accordingly to the output of data filtering
operations.





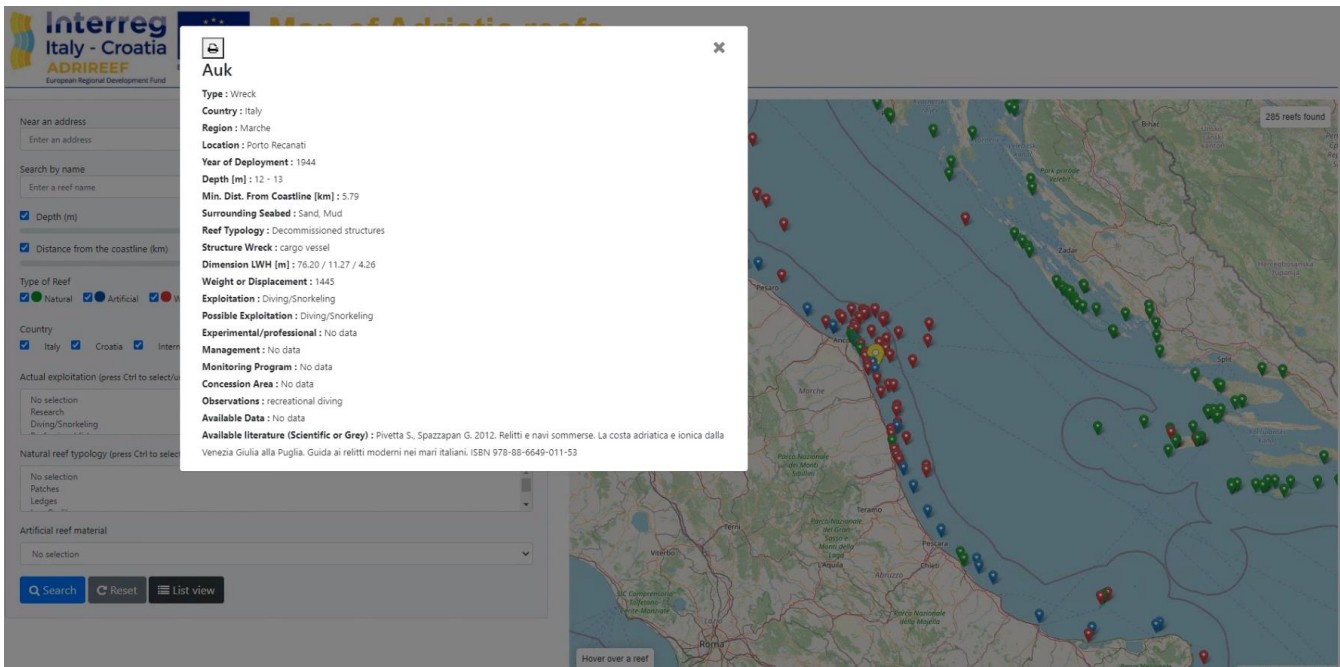


**Figure 2: Example of a pop up that appears once an element is clicked. Basemap credits: © OpenStreetMap contributors 2020.**
**Distributed under a Creative Commons BY-SA License**


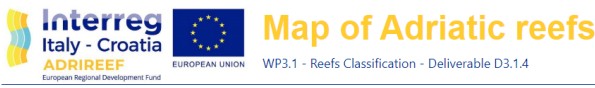

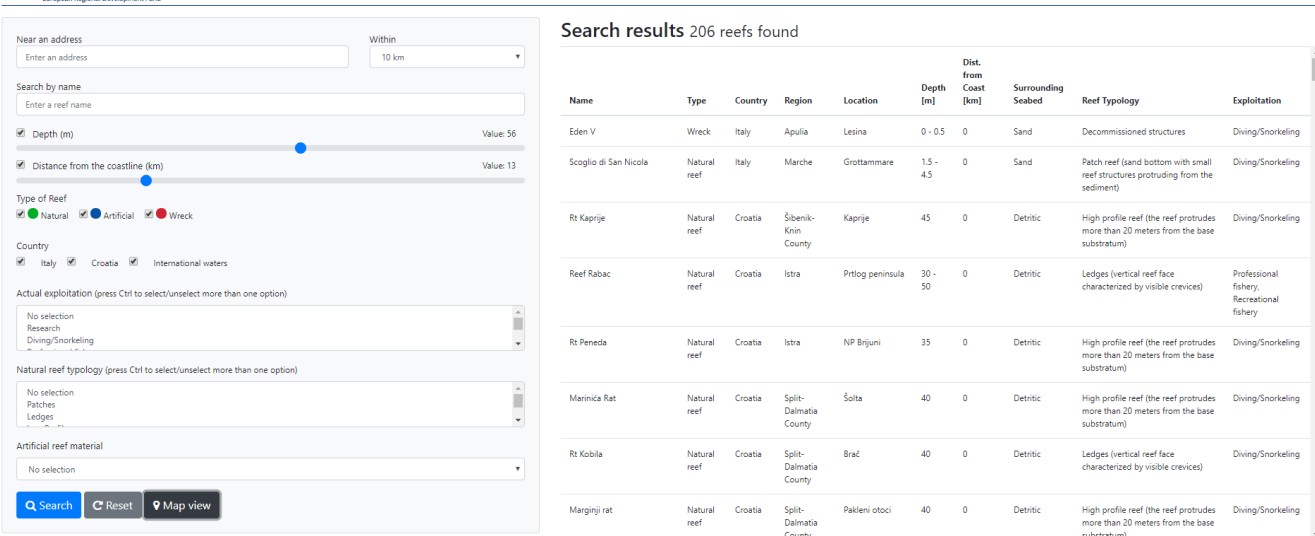


**Figure 3: List view. Once elements are selected, it is possible to obtain some common information by clicking the appropriate button.**




From a technical point of view, data were entered in .csv format, then transformed in JSON objects as "Collection of Features"
class, with prototype (generic) and object (specific) capabilities. For each element, an integer and consecutive identification
number was assigned by default. The interactive map has been published using the GitHub Pages extension
(https://pages.github.com/), which represents an easy and rapid way to make information soon available online. The base map
coming from the open-source cooperative geographical project Open Street Map (https://www.openstreetmap.org/) and the
Nominatim package for geocoding operations (https://nominatim.openstreetmap.org/) were used. The whole infrastructure is
based on Searchable Map Template – CSV project (https://github.com/datamade/searchable-map-template-csv).
**5 Data analysis**
Data contained in the database can be analysed in many different ways and for different purposes. For example, Figure 4,
representing the wrecks' sink and the artificial reefs' deployment on time (excluding missing information elements), shows
that until the 2000s the majority of artificial structures existing in the Adriatic Sea were represented by wrecks, most of which
accidentally sunken. Afterwards, almost all the manmade structures deployed on the seabed were purposely constructed
artificial reefs.

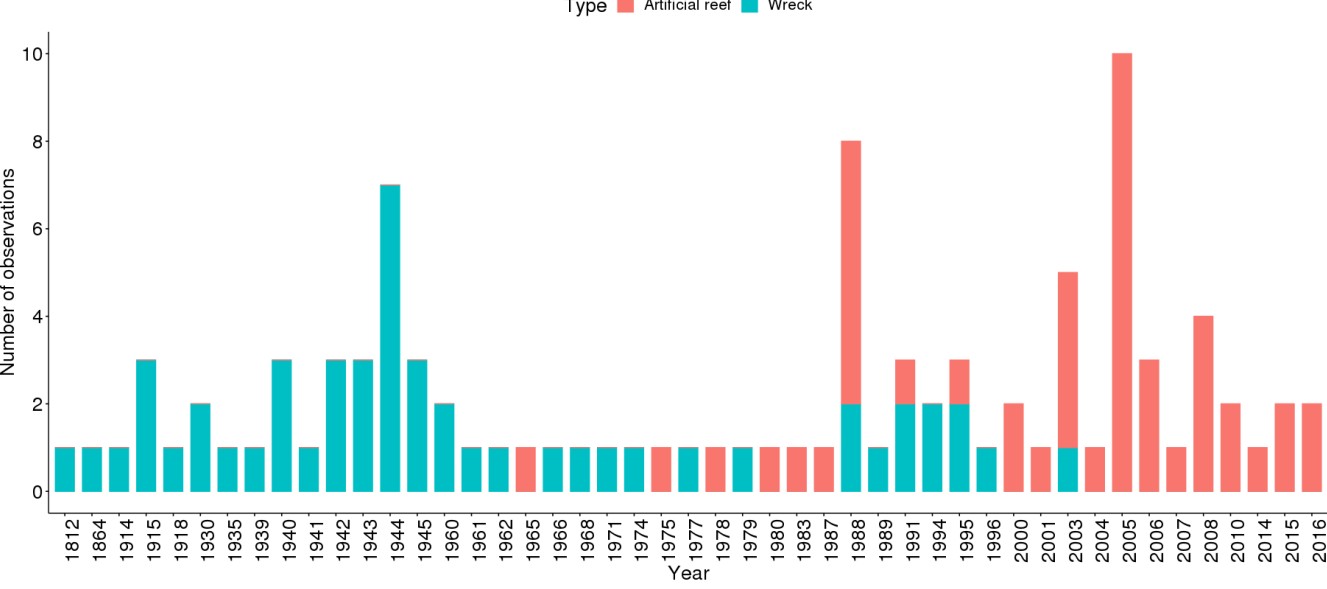


**Figure 4. Artificial reefs and wrecks by year of deployment.**



Another interesting example of analysis that can be performed on data is the evaluation of the number of natural reefs subjected
to any form of protection. Again, after deduction of "No data", it is possible to identify 31 Natura 2000 sites, 12 Protected
areas, 4 Natural reserves and 11 National parks (Fig. 5) with an overlapping at some sites. Figure 5 also shows how NRs are
distributed in these categories through Italian and Croatian waters. Coupling this graph with the information contained in Table
1, it results that almost the totality of the Italian natural reefs is subjected to some form of protection, while only half of the
Croatian ones are under preservation constraints.
Also, the 3D representation of the sites can be useful for divulgation purposes. For example, in Figure 6 (A) all reefs and
wrecks are reported over a bathymetric map of the Adriatic Sea. A section of the Tremiti Islands (Apulia, Italy), connecting
three observations of the database is reported in the zoom (B).

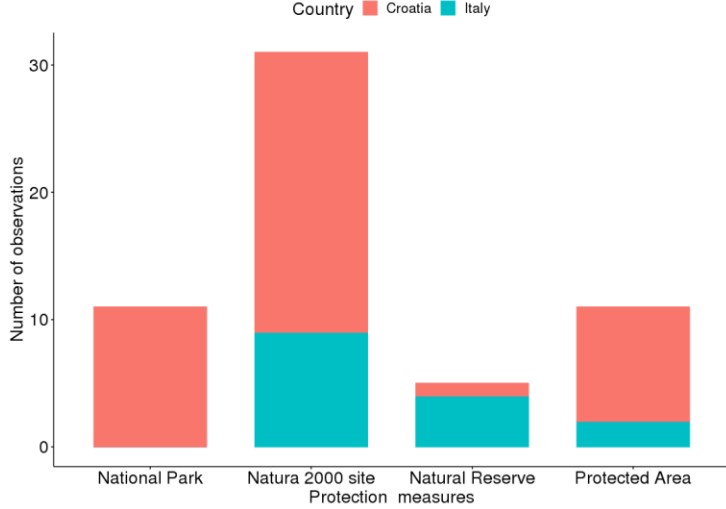

**Figure 5. Protection measures insisting on Natural Reefs both in Croatia and in Italy.**



Earth System
**Science
Data**
Open Access      Discussions

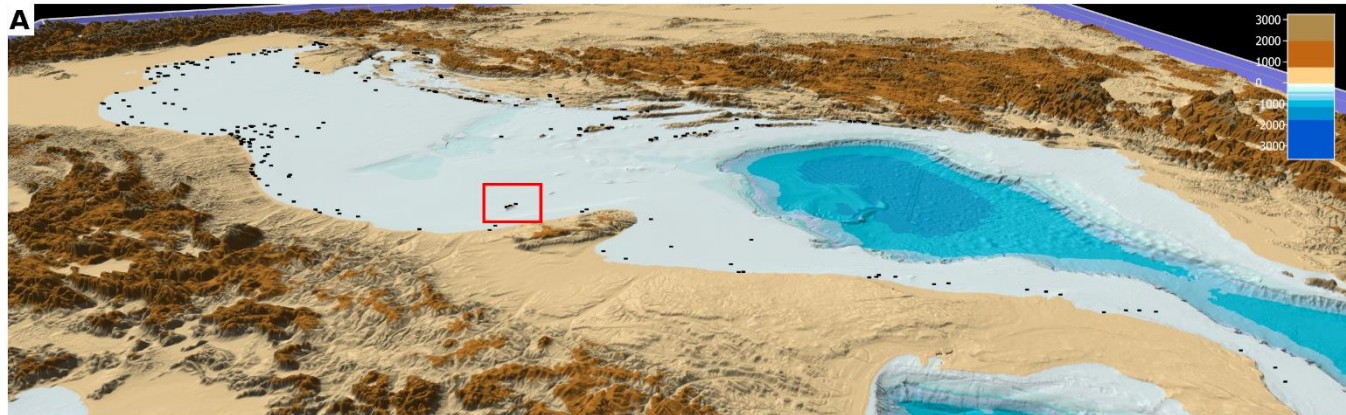

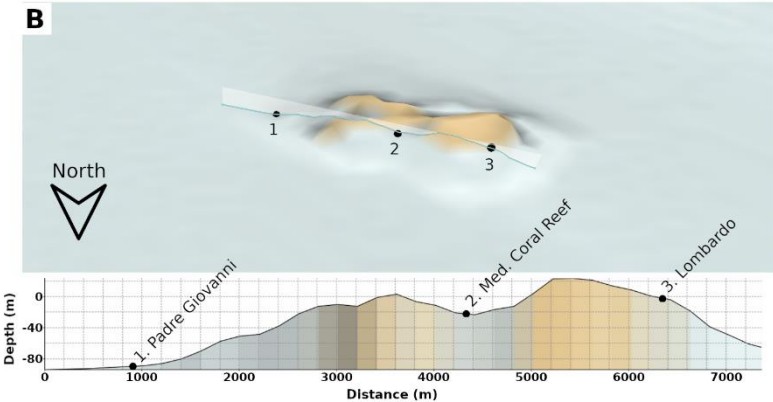

**Figure 6. A 3D visualization of the spatial distribution of natural reefs, artificial reefs and wrecks in the Adriatic Sea (A). In the red square, the area concerning the zoom reported below. A view of Tremiti Islands and the vertical terrain section from left to right side crossing three elements (B): Padre Giovanni and Lombardo wrecks (points no. 1 and 3) and Mediterranean Mesophotic Coral Reef (point no. 2).**

**6 Data availability**

The database (Ferrà et al., 2020) is currently available for download from EMODnet (European Marine Observation Data network, Novellino et al., 2015) through the SEANOE (https://www.seanoe.org/) repository and it is reachable at the following address: https://doi.org/10.17882/74880. EMODnet was chosen for two main reasons: it ensures long-term data availability and has increasingly become a reference point for all available European marine data (Martín Míguez et al., 2019). In fact, the platform was financed in the framework of EU's Integrated Maritime Policy definition (Commission of the European Community, 2007) to unlock existing but fragmented and hidden marine data and make them freely accessible for a wide range of users (Calewaert et al., 2016), while respecting FAIR data management principles (Findable, Accessible, Interoperable, Reusable; Wilkinson et al., 2016). In this way, an invaluable heritage of marine data was collected and all data uploaded in EMODnet are indexed in Web Of Science. The database was released under Creative Commons Attribution license (CC-BY, v. 4.0, https://creativecommons.org/licenses/by/4.0/deed.it)

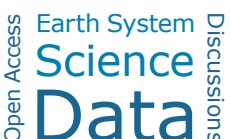

## 7 Conclusions

The data collection work and publication represent an unprecedented, consistent and robust recognition of the reefs and wrecks in the Adriatic Sea. The database fully accomplished the purposes for which it was developed as it represents a comprehensive collection providing a well-detailed state of the art and some hints on possible/future exploitation of reefs and wrecks in this geographical zone.

Indeed, the collected information can be useful for different purposes, from spatial management, to the strengthening of some economic activities and/or development of new ones taking into account the local environmental features.

Knowing the environmental status and current exploitation level of reefs located in a specific geographical area is in fact fundamental to identify potential additional ecosystem services that those reefs can provide and, consequently, develop sustainable economic activities with subsequent positive impacts on the local communities (Costanza et al., 2014). In addition, from the research point of view, a comprehensive database like the one presented here could be a starting point for the implementation of ecological studies where the information is still scarce or lacking as well as of monitoring programmes aimed at evaluating the impact of some economic activities (e.g., tourism) on sensitive habitats.

Lastly, the interactive map represents a tool that allows, through the simultaneous usage of different filters, to highlight and quantify particularly interesting situations in a user-friendly and quick manner, so to be also easily handled by the wide public. It could be, for example, used by tourists to identify suitable and less known sites for recreational activities such as snorkelling, diving and sailing.

In the overall, the provided collection can be helpful to increase visibility and attractiveness of reefs and wrecks existing in the Adriatic Sea while increasing awareness of both policy makers and citizens towards the need of managing and exploiting these sites in a sustainable way in order to assure their preservation over time.

The general perception derived from an overall evaluation of the collected data is that, in the Adriatic context, reefs and wrecks still represent an underestimated environmental heritage that, if adequately preserved and promoted, could provide in the near future new opportunities for developing activities in line with the Blue Economy.





# Appendix A: Natural Reefs Questionnaire

1.   Name and Surname: ...............................................................................................................................
2.   Occupational qualification and workplace: ............................................................................................
3.   Name of the reef: ....................................................................................................................................
4.   Location of the reef: ...............................................................................................................................
5.   Geographical coordinates Latitude (WGS84 DD.DD. e.g. 43.023N): ....................................................
6.   Geographical coordinates Longitude (WGS84 DD.DD. e.g. 13.123N): ..................................................
7.   Reef bottom depth (m) (If it is in a range, please specify the max and min): ........................................
8.   Reef edge (m): .......................................................................................................................................
9.   Minimum distance from the coast (km): ..................................................................................................
10.  Total area occupied by the Natural Reef ($m^2$): .....................................................................................
11.  Typology of the reef:

☐ High profile reef (the reef protrudes more than 20 meters from the base substratum)

☐ Low profile reef (the reef protrudes less than 20 meters from the base substratum)

☐ Ledges (vertical reef face characterized by visible crevices)

☐ Boulder reef (structure elevating from the flat seabed)

☐ Patch reef (sand bottom with small reef structures protruding from the sediment)

☐ I don't know

12.  Origin of the reef:

☐ Biogenic

☐ Geogenic

13.  Type of surrounding seabed:

☐ Rocks

☐ Sand

☐ Mud

☐ Detritic

☐ Gravel

☐ Other

14.  Occurrence of meadows?

☐ Yes, phanerogams

☐ Yes, algae

☐ no

15.  Which are the most important biocenoses? ............................................................................................
16.  Any alien species?

☐ Yes

☐ No

☐ Maybe

17.  If "Yes", which alien species? ...............................................................................................................
18.  Any protected species? (e.g. IUCN Red List of Threatened Species, ASPIM Protocol, Berna Convention, etc.)

☐ Yes

☐ No

☐ Maybe

☐ If "Yes", which species?



19. Is the natural reef within a protected area?
☐ Yes, MPA
☐ Yes, Natura 2000 site
☐ Yes, National park
☐ Yes, Natural park
☐ Yes, Marine reserve
☐ No
20. Is the reef managed?
☐ Yes
☐ No
☐ I don't know
21. If yes, which is the Managing Subject? (Please give a short summary of the management measures adopted)
22. Does exist a monitoring program?
☐ Yes
☐ No
23. If "Yes" please give a short summary of the program .......................................................................................
24. Surveillance service?
☐ Yes
☐ No
25. Current use of the Reef:
☐ Diving
☐ Mariculture
☐ Research
☐ Professional fishery
☐ Recreation fishery
☐ Fishing tourism
☐ Nothing
☐ Other (please, specify) ..............................................................................................................................
26. Development perspectives of the Natural Reef:
☐ Diving
☐ Mariculture
☐ Research
☐ Professional fishery
☐ Recreation fishery
☐ Fishing tourism
☐ Nothing
☐ Other (please, specify) ..............................................................................................................................
27. Please list the available data (If "Other" please specify):
☐ Geophysical map
☐ Water column
☐ Sediments
☐ Benthic community
☐ Fish community
☐ Other (please, specify) ..............................................................................................................................
28. Available literature (Scientific or Grey):
(Please add as many papers/works you know about the reef using the scheme:
1 Title/ 2 Authors / 3 Year of publication / 4 Journal or project / 5 Pages / 6 Abstract / 7 Keywords)

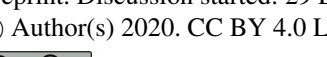



# Appendix B: Artificial Reefs Questionnaire


1.   Name and Surname: ................................................................................................................
2.   Occupational qualification and workplace: ...........................................................................
3.   Name of the reef: ..................................................................................................................
4.   Location of the reef: .............................................................................................................
5.   Geographical coordinates Latitude (WGS84 DD.DD. e.g. 43.023N): .................................
6.   Geographical coordinates Longitude (WGS84 DD.DD. e.g. 13.123N): ...............................
7.   Year of deployment of the AR: .............................................................................................
8.   Year of modification of the AR: ...........................................................................................
9.   Bottom depth (m) (If it is in a range, please specify the max and min): ...............................
10.   Minimum distance from the coast (km): ...............................................................................
11.   Type of surrounding seabed:
☐ Rocks
☐ Sand
☐ Mud
☐ Detritic
☐ Gravel
☐ Other (please, specify) ..............................................................................................
12.   Occurrence of meadows?
☐ Yes, phanerogams
☐ Yes, algae
☐ No

**ARTIFICIAL REEF STRUCTURE**
1.   Reef typology:
☐ Specifically designed modules (basic module)
☐ Decommissioned structures
☐ Other (please, specify) ..............................................................................................
**Specifically designed modules (basic module)**
1.   Material:
☐ Concrete
☐ Sea-friendly concrete (e.g., Tecnoreef)
☐ Coal Ash
☐ Rocks
☐ Fiberglass
☐ Other (please specify) ...............................................................................................
2.   Shape of the single module:
☐ Cube
☐ Pole
☐ Plinth
☐ Other (please, specify) ..............................................................................................
3.   Dimension of the single module (m): ...................................................................................
4.   Total volume of deployed material ($m^3$): ............................................................................
5.   Arrangement of the modules:
☐ Geometrically assembled to form structures
☐ Scattered



☐ Other (please, specify) ................................................................................................................

Artificial Reef geometrically assembled to form structures
1.   Typology (e.g., pyramid): ......................................................................................................
2.   Number of deployed structures: .........................................................................................
3.   Height of the structures (m): .............................................................................................
4.   Distance among structures (m): .........................................................................................
Scattered Artificial Reef
1.   Number of deployed structures: .........................................................................................
2.   Distance between structures (m): .......................................................................................
If the Artificial Reef is composed by areas or oases, please indicate:
1.   Number of the oases: .........................................................................................................
2.   Distance among oases: .......................................................................................................
3.   Dimension of each oasis ($m^2$): ........................................................................................
4.   Total area occupied by the Artificial Reef (including the area covered by the bodies, the distance between the bodies

and the area of respect) ($m^2$): ...........................................................................................

**Decommissioned structures**
Please specify the nature of the structure:
☐ Offshore extraction platform
☐ Purposely sunk vessel/ship
☐ Other (please, specify) ........................................................................................................
Offshore extraction platform:
1.   Type of the platform (e.g., one-leg platform): ...................................................................
2.   Part of the platform used to realize the AR (e.g., jacket, deck): .......................................
3.   Total area occupied by the Artificial Reef ($m^2$): ............................................................
Purposely sunk vessel/ship:
1.   Number of sunk vessels: ...................................................................................................
2.   Vessel material:

☐ Wood

☐ Iron

☐ Fiberglass

☐ Other (please, specify) ..............................................................................................

3.   Dimension of the sunk vessel/ship - LFT (m) and Weight (ton): .......................................
Other Artificial Reefs:
1.   Number of bodies: .............................................................................................................
2.   Material of bodies:

☐ Wood

☐ Iron

☐ Fiberglass

☐ Concrete

☐ Other (please, specify) ..............................................................................................

3.   Dimension of each body - length (m) and Weight (ton): ..................................................

**ARTIFICIAL REEF UTILIZATION**

1.   Scope:

☐ Habitat protection

☐ Habitat restoration

☐ Finfish enhancement





| | | |
|---|---|---|
| 474 | | ☐ Diving |
| 475 | | ☐ Mariculture |
| 476 | | ☐ Research |
| 477 | | ☐ Professional fishery |
| 478 | | ☐ Recreational fishery |
| 479 | | ☐ Fishing tourism |
| 480 | | ☐ Other (please, specify) ............................................................................................................ |
| 481 | 2. | Type of Artificial Reef: |
| 482 | | ☐ Experimental |
| 483 | | ☐ Professional |
| 484 | 3. | Is the Reef exploited at present? |
| 485 | | ☐ Yes |
| 486 | | ☐ No |
| 487 | | ☐ Maybe |
| 488 | | If "Yes", by whom? ........................................................................................................................ |
| 489 | 4. | Does exist a management program? |
| 490 | | ☐ Yes |
| 491 | | ☐ No |
| 492 | | ☐ Maybe |
| 493 | | If "Yes", please specify the Managing Subject and give a short summary of the adopted management measures |
| 494 | | ..................................................................................................................................................... |
| 495 | 5. | Concession area? |
| 496 | | ☐ Yes |
| 497 | | ☐ No |
| 498 | 6. | Surveillance service? |
| 499 | | ☐ Yes |
| 500 | | ☐ No |
| 501 | 7. | Does exist a monitoring program? |
| 502 | | ☐ Yes |
| 503 | | ☐ No |
| 504 | | ☐ Maybe |
| 505 | | If "Yes", please give a short summary (Duration / Monitored aspects / Involved Institute or Agency /address, e-mail |
| 506 | | address) ........................................................................................................................................ |
| 507 | 8. | Possible exploitation of the Artificial Reef: |
| 508 | | ☐ Diving |
| 509 | | ☐ Mariculture |
| 510 | | ☐ Research |
| 511 | | ☐ Professional fishery |
| 512 | | ☐ Recreational fishery |
| 513 | | ☐ Fishing tourism |
| 514 | | ☐ Nothing |
| 515 | | ☐ Other (please, specify) ............................................................................................................ |
| 516 | 9. | Please list the available data: |
| 517 | | ☐ Geophysical map |
| 518 | | ☐ Water column |
| 519 | | ☐ Sediments |
| 520 | | ☐ Benthic community |
| 521 | | ☐ Fish community |
| 522 | | ☐ Other (please, specify) ............................................................................................................ |



10. Available literature (Scientific or Grey):
(Please add as many papers/works you know about the reef using the scheme: 1 Title/ 2 Authors / 3 Year of
publication / 4 Journal or project / 5 Pages / 6 Abstract / 7 Keywords)
526





# Appendix C: Wreck Questionnaire

1. Name and Surname: ....................................................................................................................................
2. Occupational qualification and workplace: ...............................................................................................
3. Name of the Wreck: ....................................................................................................................................
4. Location of the Wreck: ...............................................................................................................................
5. Geographical coordinates Latitude (WGS84 DD.DD. e.g. 43.023N): ......................................................
6. Geographical coordinates Longitude (WGS84 DD.DD. e.g. 13.123N): ...................................................
7. Year of accidental sinking of the Wreck: ..................................................................................................
8. Bottom depth (m) (If it is in a range, please specify the max and min): ...................................................
9. Minimum distance from the coast (km): ....................................................................................................
10. Type of surrounding seabed:
    ☐ Rocks
    ☐ Sand
    ☐ Mud
    ☐ Detritic
    ☐ Gravel
    ☐ Other (please, specify) .........................................................................................................................
11. Occurrence of meadows?
    ☐ Yes, phanerogams
    ☐ Yes, algae
    ☐ No
12. Vessel material:
    ☐ Wood
    ☐ Iron
    ☐ Fiberglass
    ☐ Other (please, specify) .........................................................................................................................
13. Total area occupied by the Wreck ($m^2$): ...............................................................................................
14. Total volume of the Wreck ($m^3$): .........................................................................................................
15. Eventual fragments of the Wreck and their spatial configuration:...........................................................
16. Dimension of the sunk vessel/ship - LFT (m) and Weight (ton): ............................................................
17. Is the Wreck exploited at present?
    ☐ Yes
    ☐ No
    ☐ Maybe
    If "Yes", by whom? .................................................................................................................................
18. Does exist a management program?
    ☐ Yes
    ☐ No
    ☐ Maybe
    If "Yes", please specify the Managing Subject and give a short summary of the adopted management measures
    .................................................................................................................................................................
19. Concession area?
    ☐ Yes
    ☐ No
20. Surveillance service?

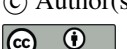



572   ☐ Yes
573   ☐ No
574   21. Does exist a monitoring program?
575   ☐ Yes
576   ☐ No
577   ☐ Maybe
578   If "Yes", please give a short summary (Duration / Monitored aspects / Involved Institute or Agency /address, e-mail
579   address) ..................................................................................................................................
580   22. Possible exploitation of the Wreck:
581   ☐ Diving
582   ☐ Mariculture
583   ☐ Research
584   ☐ Professional fishery
585   ☐ Recreational fishery
586   ☐ Fishing tourism
587   ☐ Nothing
588   ☐ Other (please, specify) ....................................................................................................
589   23. Please list the available data:
590   ☐ Geophysical map
591   ☐ Water column
592   ☐ Sediments
593   ☐ Benthic community
594   ☐ Fish community
595   ☐ Other (please, specify) ....................................................................................................
596   24. Available literature (Scientific or Grey):
597   (Please add as many papers/works you know about the reef using the scheme: 1 Title/ 2 Authors / 3 Year of
598   publication / 4 Journal or project / 5 Pages / 6 Abstract / 7 Keywords)
599



**Author contribution**

AM, CF, GF and ANT worked to the conceptualization of the paper; AM and CF analysed the data and wrote the original draft; AM, ANT, GF, AS, MSc, MSo, DB reviewed and edited the manuscript; CF, AS, MSc, ANT, GF, CRF, CM, SP, ZJ, TŠ, MŠ, CK, DP, EB, MDG, DB, EG, RA, IB, ĐVS, SO, VF, DZ, IOK, MSo, SU contributed to the collection and curation of data descripted in this paper; AM worked at data visualization; GF supervised the whole work.

**Competing interests**

Authors declare that they have no conflict of interest.

**Funding**

The present work was funded by the European Regional Development Fund in the frame of the project ADRIREEF (Innovative exploitation of Adriatic Reefs in order to strengthen blue economy), project ID: 10045901. Interreg VA Italy Croatia Cross-border Cooperation Programme 2014-2020.

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
