# Peer review of "The ADRIREEF database: a comprehensive collection of"

_Earth System Science Data, 2020_

## Referee Comment (RC1) · Chiara Venier (Referee) · 1 Feb 2021

I consider this article appropriate to support the publication of the dataset presented. So far, a comprehensive dataset of natural/artificial reefs and wrecks in the whole Adriatic Sea was missing. The possibility to use a complete mapping of this data is definitely important in the Adriatic Sea study area, in a context driven by the EUSAIR strategy (European Commission, 2014 - http://www.adriatic-ionian.eu/component/edocman/34-action-plan-eusair-pdf) and the wider Blue Growth initiative (European Commission, 2017). The importance of this study and the dataset produced, is clearly confirmed by the fact that it has been produced within the

[Figure]

ADRIREEF project, funded by the EU Interreg Italia-Croatia Programme. For these reasons, the data have potential for reuse in the future for the blue economy purposes, including the maritime spatial planning implementation. For example, the dataset can be very useful for the initial assessment and for the subsequent identification of planning measures for the sustainable development of tourism in a specific study area. Indeed, the data presented here (natural/artificial reefs and wrecks) have been already partially mapped in the Tools4MSP Data Portal (http://data.tools4msp.eu/) and PORTODIMARE (https://www.portodimare.eu/) Data Portal within the Maritime Spatial Planning framework. The datasets presented here is consistent, in terms of geographical location and other physical-ecological-economic characteristics, with the datasets present in the above-mentioned Data Portals. The methodology presented in the article, including the production and elaboration of questionnaires, is certainly an effective method to collect the most appropriate data, as well as onerous in terms of the amount of work needed for the production-identification of the addressees-the consequent data elaboration, for the creation of such a specific dataset, where the information is generally not easily reachable. The data set publication is of high quality: data are usable in the format and size provided. The most appropriate metadata have been here identified as well as described clearly and in detail. It is also clear reported where information is missing. Overall, the article is also well structured and clear, as well as language consistent and precise.

Few specific comments: 1. The methods and materials are overall described in detail in the paper. I only suggest including a clearer definition of what natural/artificial reefs and wrecks are, just at the beginning of the 2.2.1 section (before line 112). The definition of wrecks is actually included at lines 83 and 140, but a very brief introduction of the distinction of the three elements could be integrated. 2. Do you investigate also the www.relitti.it database? This database was suggested by stakeholders within the marine tourism sector (in the frame of an H2020 project) and might be compared to these wrecks data. 3. How many questionnaires were collected? only by Partners or also a wider network? This information could be included in line 152. 4. Line 183: the

"groups" are mentioned here for the first time after line 31 in the abstract, so a short introduction can be included.

Few minor technical corrections: Line 43: missing EU Directive 2014/89/UE reference Line 53: missing Pivetta et al., 2012 reference Line 60: suggestion to include the website citation of the "Reefbase project" Line 63: please correct Tara et al. with Tora et al. Line 87: please delete the point at the beginning of the line Line 91: the link is not active anymore Line 97: the website is currently unavailable Line 150: impossible with possible?
* * *

---

## Author Comment (AC1) · 5 Feb 2021

Specific comments:

1) We will insert the definition of natural reef, artificial reef, and wreck in the introduction.

2) We consulted the www.relitti.it website and we found that much information (included coordinates) was not directly available on the website but complex inquiries were required to acquire the data. Moreover, the website was offline during a consistent part of the data collection phase.

[Figure]

3) We shared the questionnaires only with project partners. The number of questionnaires collected will be included in the final version.

4) Groups' specifications will be added in paragraph 2.2.1.

Technical comments:

Line 43 and 53: we will insert the missing citations.

Line 60: we will include the Reefbase website URL, as requested.

Line 63: we will correct the citation, as suggested.

Line 91: the hyperlink of the PDF was copy-pasting a bad address mixing it with the line numbers, please make sure to copy-paste the correct spelled address.

Line 97: the URL was misspelled, we will correct it.

Line 150: will be rephrased in order to accomplish the reviewer request.

---

## Referee Comment (RC2) · Sarretta Alessandro (Referee) · 21 Feb 2021

**General comments**

The article presents an already available well-structured and relevant dataset of natural/artificial reefs and wrecks in the Adriatic Sea. The content of the dataset is presented with good detail and providing various different perspectives on possible reuse. The availability of a webGIS interface for visualizing and search the information based on several filters is also a valuable tool for both general users and stakeholders. As a general comments, the dataset refers to reefs and wrecks pertaining Italy and Croatia, while both in the title and in the rest of the article it is always referred to the Adriatic

[Figure]

Sea. The fact that information related to Slovenia, Bosnia and Herzegovina, Montene-gro and Albania is not included is somehow implicit, given that the project is an Interreg Italy-Croatia; but it should be explicitly mentioned somewhere at the beginning so that readers can clearly understand "Adriatic Sea" as "Italy+Croatia" in the context of the whole paper.

**Specific comments**

* Section 2.1 describes a literature and data review: it would be valuable to have these elements as a supplemental information to this paper

* In section 2.2.1, it would be useful to explain more clearly that the 4 questions have been used to categorize reefs in 4 groups and refer to table 2

* In table 2, it would seems more clear to have "Applicability" instead of "Applicability restriction" and simply list the type of elements were the information applies, e.g.: AR; NRs, ARs; ARs, wrecks; ...; all (or explicitly NRs, ARs, wrecks)

* In fig. 4, the fact that missing years are not represented is somehow misleading. I think a different representation would help to better communicate the deployment frequency to readers.

* In the section 6 Data availability (and in a few other places in the manuscript) it is men-tioned that the database is available from EMODnet, and then referring to the SEANOE repository. From the documentation (https://www.seanoe.org/html/about.htm) I have understood that SEANOE duplicates records from its repository into the EMODnet Data Ingestion portal but it's not clear whether the specific dataset, described here, has been already included in any of the EMODnet portals/catalogues. Could you please clarify this and update the manuscript accordingly?

* In the webGIS interface (https://adrireef.github.io/sandbox3/) there seems to be no explicit way to download the filtered elements after a specific search. This could be a useful added functionality for users. In addition to that, it would be good to have an explicit reference in the webGIS interface both to the original dataset (http://doi.org/10.17882/74880) and the data paper (https://essd.copernicus.org/preprints/essd-2020-384/) to allow interested users to check sources, methods an references.

**Technical corrections See attached document including comments and suggested corrections.**

Please also note the supplement to this comment:
https://essd.copernicus.org/preprints/essd-2020-384/essd-2020-384-RC2-supplement.pdf

**Supplement:**

[revised manuscript text omitted]

---

## Author Comment (AC2) · 9 Mar 2021

General comments:

We specified that the study regards the portion of the Adriatic Sea that is composed of Italian, Croatian and International waters in the introduction.

Specific comments:

Q1: Section 2.1 describes a literature and data review: it would be valuable to have these elements as a supplemental information to this paper A1: The bibliography functional to the dataset enrichment has been released contextually with the revised version

of the data paper as supplemental material. All the consulted literature is also available for download with the data from SEANOE repository. More specifically the .xlsx version of the database has a tab called "Literature" reporting for each reef the consulted material (in a relation of "one to many").

Q2: In section 2.2.1, it would be useful to explain more clearly that the 4 questions have been used to categorize reefs in 4 groups and refer to table 2 A2: The four main groups of questions have been evidenced in section 2.2.1, as required.

Q3: In table 2, it would seems more clear to have "Applicability" instead of "Applicability restriction" and simply list the type of elements were the information applies, e.g.: AR; NRs, ARs; ARs, wrecks; ...; all (or explicitly NRs, ARs, wrecks) A3: The field "Applicability" in Table 2 has been changed accordingly to the suggestions of the reviewer.

Q4: In fig. 4, the fact that missing years are not represented is somehow misleading. I think a different representation would help to better communicate the deployment frequency to readers. A4: Figure 4 has been corrected including periods where no ARs were deployed and no wrecks sunk, as demanded.

Q5: In the section 6 Data availability (and in a few other places in the manuscript) it is mentioned that the database is available from EMODnet, and then referring to the SEA-NOE repository. From the documentation (https://www.seanoe.org/html/about.htm) I have understood that SEANOE duplicates records from its repository into the EMOD-net Data Ingestion portal but it's not clear whether the specific dataset, described here, has been already included in any of the EMODnet portals/catalogues. Could you please clarify this and update the manuscript accordingly? A5: The database is currently undergoing the appropriate procedure to be published in the EMODnet catalogs, but it is still in Data Ingestion Portal. We corrected as requested in the appropriate sections.

Q6: In the webGIS interface (https://adrireef.github.io/sandbox3/) there seems to be no explicit way to download the filtered elements after a specific

search. This could be a useful added functionality for users. In addition to that, it would be good to have an explicit reference in the webGIS interface both to the original dataset (http://doi.org/10.17882/74880) and the data paper (https://essd.copernicus.org/preprints/essd-2020-384/) to allow interested users to check sources, methods and references. A6: The WebGIS has been conceived only as a viewer from the original purpose. For sure all the suggested modifications would be valuable for the users and we will consider implementing them. The references to data and the data paper have been added to the WebGIS interface.

Technical correction:

Corrections and suggestions indicated in the supplemental PDF have been incorporated in the revised version of the manuscript.

Please also note the supplement to this comment:
https://essd.copernicus.org/preprints/essd-2020-384/essd-2020-384-AC2-supplement.pdf

[Figure]

**Fig. 1.** Artificial reefs and wrecks by year of deployment at decadal scale.

**Supplement:**

**Available Literature (scientific or gray):**

2007. Le barriere artificiali ADRI.BLU in Veneto. AT 4 -Azioni pilota finalizzate all'incremento della sostenibilità delle attività di pesca. Gestione sostenibile delle attività di pesca e delle risorse alieutiche dell'Adriatico. 2007. Progetto ADRIBLU. INTERREG III A - Alto Adriatico. p: 106-114.

2016. WP 4.3 - Incremento della biodiversità in Regione del Veneto - Reefball. Progetto EcoSea - Protezione, miglioramento e gestione integrata dell'ambiente marino e delle risorse naturali transfrontaliere. Programma Operativo Ipa Adriatico 2007-2013. p: 70-72.

2016. WP 4.3 - Incremento della biodiversità in Regione Friuli Venezia Giulia - Dispositivi di Concentrazione Ittica. Progetto EcoSea - Protezione, miglioramento e gestione integrata dell'ambiente marino e delle risorse naturali transfrontaliere. Programma operativo Ipa Adriatico 2007-2013. p: 63-66.

A.A.V.V. - 2003- Sottoprogetto ""Oasi sottomarina"" Dosso di Santa Croce - fase II - Fondazione CRTrieste.

AA.VV. Relazione finale del Progetto adri.blu Interreg IIIA alto adriatico. Gestione sostenibile delle attività di pesca e delle risorse alieutiche dell'Adriatico. 157 pp. http://www.altoadriatico.com/altoadriatico.com/doc8e72.html?iddoc=9&idarea=4

Antonini C., 2007. La Zone di Tutela Biologica di Chioggia: stima dei popolamenti ittici e aspetti della biologia di una specie commerciale Diplodus annularis (teleostei, Sparidae). Tesi di Laurea in Biologia marina, Univ. Padova, Rel. Dott.ssa Mazzoldi, 125 pp.

Arbuatti A. 2015. Studio qualitativo della fauna subacquea associata alle barriere artificiali sommerse nella costa dei Trabocchi (CH) mediante underwater visual census. 1° congresso internazionale SIVAE, congresso SCIVAC.

Arbuatti A. 2015. Studio qualitativo mediante UVC della fauna acquatica associata alle barriere artificiali sommerse nella costa dei Trabocchi (CH). Atti del convegno La medicina Veterinaria nell'Abruzzo wild. Teramo,19-20 marzo 2015.

ARPAV e Fondazione Musei Civici Venezia. 2010. Le tegnùe dell'Alto Adriatico: valorizzazione della risorsa marina attraverso lo studio di aree di pregio ambientale. pp: 203.

Association Sunce field mapping.

Auriemma R. 2007. Studio dell'evoluzione dei popolamenti bentonici del dosso di S. Croce (Golfo di Trieste, Alto Adriatico) in seguito alla posa in opera di strutture artificiali sommerse"". PhD thesis. 104 pp.

Bertasi E., 2007. Distribuzione spaziale e variazione temporale di stadi planctonici di invertebrati sulle Tegnùe di Chioggia. Tesi di Laurea specialistica in Biologia Marina, Univ. Padova, Rel. Dott.ssa Bressan, 90 pp.

Bianchi F., 2006. Progetto integrato "Tegnùe". Primi dati idrologici. Incontro annuale ass. Tegnùe di Chioggia Sottomarina di Chioggia. Dicembre 2006. Poster.

Bisca A., Ricci V.G., Pepoli R., Rambelli F., Vistoli G.P. 1994. Paguro. Immagini da un relitto. Edizioni Calderini, Bologna. 152 pp.

Bombace G. 1973. Progetto per la realizzazione di barriere artificiali nel mare di Ancona. Gazzettino della Pesca, 20 (1).

Bombace G. 1977. Aspetti teorici e sperimentali concernenti le barriere artificiali. Atti del IX Congresso della Società Italiana di Biologia Marina: 29-41 pp.

Bombace G. 1978. Le scogliere del Conero. Il Subacqueo, 66-68 pp.

Bombace G. 1979. Esperienze di creazione di barriere artificiali in Medio Adriatico (S.E. Conero-Ancona). Atti del Convegno Nazionale P.F. Oceanografia e Fondi Marini, Roma 5-7 marzo 1979: 185-198 pp.

Bombace G. 1979. Nota sull'esperimento di barriere artificiali a fini multipli realizzato dal laboratorio di tecnologia della pesca del C.N.R. - Ancona. Gazzettino della Pesca: 1-8 pp.

Bombace G. 1980. Paper on experiments on artificial reefs in Italy. Simposium on management of living resources in the Mediterranena coastal area, CGPM, Palma de Majorca, 18-20 September 1980.GFCM/XV/80/21: 1-112 pp.

Bombace G. 1981. Note on experiments in artificial reefs in Italy. Symposium on management of living resources in the Mediterranean coastal area. FAO-GFCM Studies and Reviews, 58: 309-327 pp.

Bombace G. 1982. Il punto sulle barriere artificiali: problemi e prospettive. Naturalista siciliano, S. IV, VI (Suppl.). 3: 573-591 pp.

Bombace G. 1987. Iniziative di protezione e valorizzazione della fascia costiera mediante barriere artificiali a fini multipli. Atti della LIX Riunione della SIPS: 201-233 pp.

Bombace G. 1989. Le ricerche nella fascia costiera base allo sviluppo della piccola pesca. Nova Thalassia 10 (1): 57-80 pp.

Bombace G. 1994. Artificial reefs in the Mediterranean sea. Bulletin of Marine Science, 44(2): 1023-1032 pp.

Bombace G. 1995. La barriere artificiali nella gestione razionale della fascia costiera italiana. Biologia Marina Mediterranea, 2(1): 1-14 pp.

Bombace G. 2016. Barriere artificiali e mitilicoltura in mare aperto. Bollettino della Società Italiana di Biologia Marina, 69: 63-71 pp.

Bombace G., Artegiani A., Fabi G., Fiorentini L. 1991. Ricerche comparative sulle condizioni ambientali e sulle possibilità biologiche e tecnologiche di allevamento ottimale di mitili ed ostriche in mare aperto mediante strutture sommerse e sospese. Rapporto al Ministero Marina Mercantile, Direzione Generale della Pesca e dell'Acquacoltura. 205 pp.

Bombace G., Castriota L., Spagnolo A. 1997. Benthic community on concrete and coal-ash blocks submerged in an artificial reef in the central Adriatic Sea. Pages 281-290 In: Hawkins L.E. et al. Proceedings of the 30th European Marine Biology Symposium, 18-20 September 1995, Southampton, UK.

Bombace G., Fabi G., Fiorentini L. 1989. Preliminary analysis of catch data from artificial reefs in Central Adriatic. General Fisheries Council for the Mediterranean. FAO, 428: 86-98 pp.

Bombace G., Fabi G., Fiorentini L. 1989. Monitoring program associated to the construction of the AR (1989 - 1991),"Benthic community, Fish community, Sediments",Preliminary analysis of catch data from artificial reefs in Central Adriatic. General Fisheries Council for the Mediterranean. FAO, 428: 86-98 pp.

Bombace G., Fabi G., Fiorentini L. 1990. Catch data from an artificial reef and a control site along the Central Adriatic coast Rapport de la Commission internationale pour l'exploration scientifique de la Méditerranée, 32 (1): 247 pp.

Bombace G., Fabi G., Fiorentini L. 1993. Aspects theoriques et resultats concernants les recifs artificiels realises en Adriatique. Bollettino di Oceanologia Teorica ed Applicata 11 (3-4): 145-154 pp.

Bombace G., Fabi G., Fiorentini L. 1995. Osservazioni sull'insediamento e l'accrescimento di Pholas dactylus L. (Bivalvia, Pholadidae) su substrati artificiali.Biologia Marina Mediterranea, 2 (2): 143-150 pp.

Bombace G., Fabi G., Fiorentini L. 2000.  Artificial Reefs in the Adriatic Sea. A.C. Jensen et al. (eds.), Artificial Reefs in European Seas, Kluwer Academic Publishers in Great Britain, 31-63 pp.

Bombace G., Fabi G., Fiorentini L., Spagnolo A. 1996. Studi ed esperimenti su: a) Iniziative di allevamento di astici e orate in mare aperto mediante gabbie, b) Sistemi ""FADs"" per l'attrazione e la concentrazione di pesci pelagici. Rapporto per Ministero Risorse Agricole, Alimentari e Forestali, Direzione Generale della Pesca e dell'Acquacoltura. 91 pp.

Bombace G., Fabi G., Fiorentini L., Spagnolo A. 1997. Assessment of the ichthyofauna of an artificial reef through visual census and trammel net: comparison between the two sampling techniques. Pagg. 291-305 In: Hawkins L.E. et al. Proceedings of the 30th European Marine Biology Symposium, 18-20 September 1995, Southampton, UK.

Bombace G., Fabi G., Fiorentini L., Speranza S. 1994. Analysis of the efficacy of artificial reefs located in five different areas of the Adriatic Sea. Bulletin of Marine Science 55 (2-3), 559-580 pp.

Bombace G., Fabi G., Gaetani G. 1996. Sperimentazione di un prototipo di gabbia da fondo per l'ingrasso di pesce in medio Adriatico. Biologia Marina Mediterranea, 3 (1): 186-191 pp.

Bombace G., Fabi G., Giorgi U. 1993. Ricerche sull'ittioplancton e sulle forme giovanili di pesci in barriere artificiali. Rapporto per Ministero Marina Mercantile, Direzione Generale della Pesca e dell'Acquacoltura. 84 pp.

Bombace G., Fabi G., Grati F. Spagnolo A. 1998. Tecnologie per l'allevamento in mare aperto di Sparus aurata in medio Adriatico. Biologia Marina Mediterranea, 5(1): 439-449 pp.

Bombace G., Fabi G., Grati F., Panfili M., Spagnolo A. 1998. Maricoltura associata a barriere artificiali. Biologia Marina Mediterranea, 5 (3): 1773-1782 pp.

Bombace G., Fabi G., Leonori J., Sala A., Spagnolo A. 1998. Valutazione con tecnica elettroacustica della biomassa vagile presente in una barriera artificiale del Medio Adriatico.Biologia Marina Mediterranea,  5 (3): 1844-1854 pp.

Bombace G., Fabi G., Sala A., Spagnolo A. 1998. Esperimenti volti alla valutazione della biomassa vagile in strutture artificiali di zone marine protette. Rapporto per Ministero per le Politiche Agricole, Direzione Generale della Pesca e dell'Acquacoltura. 79 pp.

Bombace G., Fabi G., Scarcella G. 2004. Valutazione dello sforzo di pesca applicabile ad una barriera artificiale del medio Adriatico. Rapporto per Ministero delle Politiche Agricole e Forestali, Dipartimento delle Politiche di Mercato, Direzione Generale per la Pesca e l'Acquacoltura. 87 pp.

Bombace G., Fabi G., Spagnolo A. 1997. Sperimentazione di strutture idonee all'allevamento di pesci in aree di mare aperto del medio Adriatico. Rapporto per Ministero Risorse Agricole, Alimentari e Forestali, Direzione Generale della Pesca e dell'Acquacoltura. 76 pp.

Bombace G., Piccinetti C. 1989. Barriere artificiali e maricoltura. Convegno Nazionale Per la difesa dell'Adriatico, Ancona 1989: 92-99 pp.

Bombace G., Rivas G., Maffei M. 1998. Valutazione dell'efficacia delle strutture artificiali nei mari italiani: valutazione degli aspetti socio-economici conseguenti alla pesca esercitata presso le barriere artificiali a fini multipli e strutture estrattive offshore del medio Adriatico. Rapporto per il Ministero delle Politiche Agricole, Direzione Generale della Pesca e dell'Acquacoltura. 62 pp.

Bombace G., Rossi V. 1986. Effets socio-economique consecutifs a la realisation d'une zone marine protegee par des recifs artificiels dans las zone de Porto Recanati. Annex (E). Roma, General Fisheries Council for the Mediterranean. FAO, 357 pp.

Boscolo S., Borromeo S., Franceschini G., Cornello M., 2006. La fauna di fondo mobile e la pressione di pesca a strascico nell'area delle Tegnùe di Chioggia (Adriatico settentrionale). Biol. Mar. Mediterr., 13(1): 566-560 pp.

Boscolo, S., Borromeo, S., Franceschini, G., Cornello, M., Giovanardi, O. 2005. La fauna di fondo mobile e la pressione di pesca a strascico nell'area delle Tegnùe di Chioggia (Adriatico settentrionale). 127. In: Riassunti del 36° Congresso nazionale della Società Italiana di Biologia Marina. Trieste. SIBM 324 pp.

Brambati A., De Muro S., Montesanti A. 1996. Studio geologico tecnico di fattibilità di una barriera sottomarina sul Dosso di Santa Croce (Golfo di Trieste). Hydrores XIII (14): 7-13 pp.

Bressan G. 1988. Appunti sulla fattibilità di una barriera artificiale sommersa nel Golfo di Trieste: processi di colonizzazione e fitocenosi guida. Hydrores, 5 (6): 47-56 pp.

Bressan G. 2006. Studio della produttività primaria e della produttività secondaria delle strutture artificiali sommerse poste in prossimità del Dosso di S. Croce (Golfo di Trieste, Alto Adriatico). EUT - Edizioni Università di Trieste. 500 pp.

Brunetti B., Spagnolo A., Punzo E., Fabi G. 2013. Effects of an artificial reef on the soft-bottom community (Western Adriatic Sea). Rapport de la Commission internationale pour l'exploration scientifique de la Méditerranée, 40: 661 pp.

Bruno S., Bastianini M., Acri F., Bon, D., Rizzardi S., Vazzoler M., Bertaggia R., Boldrin A. 2006. Idrologia e idrodinamica in un'area a barriere artificiali del Nord Adriatico. Biologia Marina Mediterranea, 13 (1): 688-692 pp.

Busatto E., 2007. Mesozooplancton dell'area delle tegnùe di Chioggia. Tesi di Laurea in Biologia Marina, Univ. Padova, Rel. Dott.ssa Bressan, 84 pp.

Camilletti E. 1998. Osservazioni sull'alimentazione di Umbrina cirrosa (L.) in una barriera artificiale del medio Adriatico. Università Politecnica delle Marche, Thesis Degree. 73 pp.

Camuffo, M., 2000. La Gestione di un bene ambientale (la tegnua di Porto Falconera al largo di Caorle) ed il problema della sua rappresentazione scientifica e vernacolare. Tesi di Laurea in Scienze Ambientali, Univ. Venezia.

Casellato S., Sichirollo E., Cristofoli A., Masiero L., Soresi S., 2005. Biodiversità delle tegnùe di Chioggia, Zona di Tutela Biologica del Mar Adriatico. Biol. Mar. Med., 12(1): 69-77 pp.

Castriota L., Fabi G., Spagnolo A. 1996. Evoluzione del popolamento bentonico insediato su substrati in calcestruzzo immersi in Medio Adriatico. Biologia Marina Mediterranea, 3 (1), 120-127 pp.

Cenci E., Mazzoldi C. 2005. Le Tegnue di Chioggia: un'analisi qualitativa e quantitativa della fauna ittica. 224. In: Riassunti del 36° Congresso nazionale della Società Italiana di Biologia Marina. Trieste. SIBM 324 pp.

Cenci E., Mazzoldi C., 2006. Le tegnùe di Chioggia: prima valutazione qualitativa e quantitativa della fauna ittica. Biol. Mar. Medit., 13(1): 840-843 pp.

Cicconi E. 1998. Regime alimentare di Diplodus annularis nella barriera artificiale di Cesano-Senigallia (medio Adriatico). Università Politecnica delle Marche, Thesis Degree. 79 pp.

Ciriaco S., Costantini M., Italiano C., Odorico R., Picciulin M., Verginella L., Spoto M., 1998. Monitoring the Miramare Marine Reserve: assessment of protection efficiency. Ital. J. Zool., 65: 383–386 pp.

Codarin A., Wysocki L.E., Ladich F., Picciulin M., 2009. Effects of ambient and boat noise on heating and communication in three fish species living in a marine protected area (Miramare, Italy). Mar. Pollut. Bull., 58(12): 1880–1887 pp.

Congi A. 2012. Vive il mare. Le barriere artificiali sommerse lungo la costa emiliano-romagnola. Centro Stampa Regionale di Bologna. 87 pp.

Congi A. 2013. Nuova vita tra le barriere artificiali sommerse. Ecoscienza, Numero 4, Anno 2013, p. 56-57.

Corriero G., Pierri C., Mercurio M., Marzano C.N., Tarantini S.O., Gravina M.F., Lisco S., Moretti M., De Giosa F., Valenzano E., Giangrande A., Mastrodonato M., Longo C., Cardone F. 2019. A Mediterranean mesophotic coral reef built by non-symbiotic scleractinians. Scientific Reports, 17 p.

Dalla Venezia L. 1979. Prime osservazioni sugli insediamenti di Mytilus sp. sulle barriere artificiali del Conero (AN). Atti Convegno Scientifico Nazionale Progetto finalizzato Oceanografia, Roma, 5-7 Marzo 1979, Vol. 2.

Dalla Venezia L. 1981. Studio comparativo di popolazioni di Mytilus galloprovincialis su substrati artificiali. Quaderni del Laboratorio di Tecnologia della Pesca, 3(1): 631-633 pp.

Danovaro R., Gambi C., Mazzola A., Mirto S. 1999. Influence of artificial reefs on the surrounding infauna: analysis of meiofauna. Proceedings Seventh International Conference on Artificial Reefs and related Aquatic Habitats (7th CARAH), Sanremo 1999: 614.

Danovaro R., Gambi C., Mazzola A., Mirto S. 2002. Influence of artificial reefs on the surrounding infauna: analysis of meiofauna. ICES Journal of Marine 59 (1 Suppl.): 356-362 pp.

Di Maio A., Marcelli M., Peviani M.A. 2004. Studio sull'assetto delle barriere artificiali sommerse destinate al ripopolamento ittico nei siti di Portonovo e Senigallia. Bollettino dell'Associazione di ingegneria offshore e marina, 29-29: 5-13 pp.

Fabi G. 1998. Fishing yield - research protocol. In: A.C. Jensen (ed) EARRN Final Report & Reccomendations. Southampton Oceanography Centre, Southampton, UK.

Fabi G. 1998. The use of artificial reefs for aquaculture in the future. In A.C. Jensen (ed) EARRN Final Report & Recommendations. Southampton Oceanography Centre, Southampton, UK.

Fabi G. 2006. Le Barriere artificiali in Italia. Pagg. 20-34 in Campo Sperimentale in mare: prime esperienze nel Veneto relative a elevazioni del fondale con materiale inerte. ARPAV, Padova.

Fabi G. 2006. Monitoring program associated to the construction of the AR (2002-2007), Benthic community, Fish community, Geophysical map, PAH in mussels, Sediment. Le Barriere artificiali in Italia. Pagg. 20-34 in Campo Sperimentale in mare: prime esperienze nel Veneto relative a elevazioni del fondale con materiale inerte. ARPAV, Padova.

Fabi G., Bolognini L., Punzo E., Spagnolo A. 2008. Monitoraggio volto alla valutazione degli effetti indotti dalla realizzazione della barriera artificiale a fini multipli Casteldimezzo - Monte Castellaro - VI anno di indagine (IV anno dopo la posa dei substrati). Rapporto per Regione Marche - Servizio Attività ittiche, Commercio e Tutela del Consumatore, Caccia e Pesca Sportiva. 75 pp.

Fabi G., Camilletti E., Cicconi E., Luccarini F., Lucchetti A., Panfili M., Solustri C. 1998. Ruolo trofico della barriera artificiale di Cesano-Senigallia nei.confronti di alcune specie ittiche. Biolgia Marina Mediterranea, 5 (1): 1812-1821 pp.

Fabi G., Ferrà C., Grati F., Guicciardi S., Pellini G., Spagnolo A., Tassetti A.N. 2017. Parte 4: Valutazione degli effetti delle Zone di Tutela Biologica (ZTB) nei mari Italiani. Convenzione tra MIPAAF e CNR-ISMAR Ancona per aggiornamenti dei piani di gestione delle specie demersali delle GSA: 9, 10, 11, 15, 16, 17, 18, 19, fermo biologico nell'area di Pomo, valutazione della pesca dei bivalvi nella fascia costiera compresa nelle 0,3 miglia nautiche e misure gestionali ZTB. 69 pp.

Fabi G., Fiorentini L. 1989. Shellfish culture associated with artificial reefs. General Fisheries Council for the Mediterranean. FAO, 428: 99-107 pp.

Fabi G., Fiorentini L. 1993. Catch and growth of Umbrina cirrosa (L.) around artificial reefs in the Adriatic Sea. Bollettino di Oceanologia Teorica e Applicata, 11 (3-4): 235-242 pp.

Fabi G., Fiorentini L. 1994. Comparison between an artificial reef and a control site in the Adriatic Sea: analysis of four years of monitoring. Bulletin of Marine Science, 55 (2-3): 538-558 pp.

Fabi G., Fiorentini L. 1997. Molluscan aquaculture on reefs. Pages 123-140 In: Jensen A.C. (Ed.) ""European Artificial Reef research"". Proceedings of the 1st EARNN Conference, Ancona, Italy, March 1996.

Fabi G., Fiorentini L., Giannini S. 1985. Osservazioni sull'insediamento e sull'accrescimento di Mytilus Galloprovincialis Lamk. su di un modulo sperimentale per mitilicoltura immerso nella Baia di Portonovo (Promontorio del Conero, Medio Adriatico). Oebalia, 11 (2): 681-692 pp.

Fabi G., Fiorentini L., Giannini S. 1986. Growth of Mytilus galloprovincialis Lamk on a suspended and immersed culture in the Bay of Portonovo (Central Adriatic Sea). Roma, General Fisheries Council for the Mediterranean. FAO, 357: 144-154 pp.

Fabi G., Fiorentini L., Giannini S. 1989. Experimental shellfish culture on artificial reefs in the Adriatic Sea. Bulletin of Marine Sciences, 44 (2): 923-933 pp.

Fabi G., Fiorentini L., Speranza S., Spagnolo A. 1996. Sperimentazione di moduli in composto cenere di carbone per la realizzazione di barriere artificiali. Relazione finale 1992-96. Rapporto per il Centro Ricerca e Valorizzazione Residui (CRR) dell'ENEL di Brindisi. 65 pp.

Fabi G., Grati F. 2005. Ripopolamento attivo di lagune,stagni costieri, e localizzate aree della fascia costiera con giovanili certificati di specie ittiche e di crostacei, secondo i principi del Codice di Condotta per una Pesca Responsabile (FAO 95) - Prove di ripopolamento in una barriera artificiale del medio Adriatico tramite immissione di esemplari di Dicentrarchus labrax. Rapporto finale per Ministero delle Politiche Agricole e Forestali, Dipartimento delle Politiche di Mercato, Direzione Generale per la Pesca e l'Acquacoltura. 68 pp + iii.

Fabi G., Grati F., Luccarini F., Panfili M. 1999. Indicazioni per la gestione razionale di una barriera artificiale: studio dell'evoluzione del popolamento necto-bentonico. Biologia Marina Mediterranea, 6 (1): 81-89 pp.

Fabi G., Luccarini F., Panfili M., Solustri C., Spagnolo A. 2002. Effects of an artificial reef on the surrounding soft-bottom community (central Adriatic Sea). ICES Journal of Marine Science, 59 (1 Suppl.): 343-349 pp.

Fabi G., Luccarini F., Panfili M., Spagnolo A. 1999. Valutazione dell'efficacia delle strutture artificiali nei mari italiani: studio del funzionamento del sistema barriere artificiali attraverso le reti trofiche. Rapporto per Ministero per le Politiche Agricole, Direzione Generale della Pesca e dell'Acquacoltura. 140 pp.

Fabi G., Luccarini F., Spagnolo A. 2000. Studio sull'insediamento e l'allevamento in mare del bivalve Pholas dactylus (Pholadidae) mediante moduli artificiali in composto-cenere. Rapporto per Ministero per le Politiche Agricole, Direzione Generale della Pesca e dell'Acquacoltura. 60 pp.

Fabi G., Manoukian S., Panfili M., Solustri C., Spagnolo A. 2003. Benthic community settled on an artificial reef in the western Adriatic Sea (Italy). Oceans 2003 MTS/IEEE, Conference Proceedings. Holland Publications, Escondido, CA: 812 (abstract).

Fabi G., Manoukian S., Pupilli A., Scarcella G., Spagnolo A. 2006. Monitoring program associated to the construction of the AR (2000-2006),Benthic community, Fish community, Geophysical map, PAH in mussels, Sediments. Monitoraggio volto alla valutazione degli effetti indotti dalla realizzazione della barriera artificiale a fini multipli P.to Recanati - P.to Potenza Picena. VI Anno di indagine (2005) - V anno dalla posa in opera della barriera artificiale. Rapporto per Regione Marche - Servizio Attività ittiche, Commercio e Tutela del Consumatore, Caccia e Pesca Sportiva. 104 pp.

Fabi G., Manoukian S., Spagnolo A. 2003. Investigating terrain changes around two artificial reefs by using the multibeam echosounder. Oceans 2003 MTS/IEEE, Conference Proceedings. Holland Publications, Escondido, CA: 813 (abstract).

Fabi G., Manoukian S., Spagnolo A. 2006. Feeding behaviour of three common fishes at an artificial reef in the Northern Adriatic Sea. Bulletin of Marine Science, 78 (1): 39-56 pp.

Fabi G., Marini M., Palladino S. (eds.) 2003. - L'area marina antistante il Pomontorio del Monte Conero. Quaderni dell'Istituto Ricerche Pesca Marittima, Ancona, Nuova Serie, 1: 139 pp.

Fabi G., Panfili M., Solustri C., Spagnolo A. 2001. Osservazioni sulla fauna bentonica rinvenuta in fori scavati da Pholas dactylus (Bivalvia, Pholadidae) in substrati artificiali. Biologia Marina Mediterranea, 8 (1): 271-274 pp.

Fabi G., Panfili M., Spagnolo A. 1998. Note on feeding of Sciaena umbra L. (Osteichthyes: Sciaenidae) in the Central Adriatic Sea. Rapport de la Commission internationale pour l'exploration scientifique de la Méditerranée, 35 (2): 426-427 pp.

Fabi G., Punzo E., Scarcella G., Spagnolo A. 2011. Allevamento del dattero bianco Pholas dactylus su substrati artificiali: fase finale di sperimentazione. Rapporto finale. Rapporto per il Ministero delle politiche agricole e forestali - Dipartimento delle filiere agricole e agroalimentari, Direzione generale della pesca marittima e dell'acquacoltura. 91 pp.

Fabi G., Sala A. 2002. An assessment of biomass and diel activity of fish at an artificial reef (Adriatic sea) using stationary hydroacoustic technique. ICES Journal of Marine Science, 59: 411-420 pp.

Fabi G., Santelli A., Scarcella G. 2009. Monitoring program associated to the construction of the AR (2005-2009), Benthic community, Fish community, Geophysical map, PAH in mussels, Sediments. Monitoraggio volto alla valutazione degli effetti indotti dalla realizzazione della barriera artificiale a fini multipli Pedaso - Cupra Marittima. V anno di indagine (2006) - IV anno dopo la posa in opera dei substrati artificiali. Rapporto finale. Rapporto per Regione Marche - Servizio Attività ittiche, Commercio e Tutela del Consumatore, Caccia e Pesca Sportiva. 93 pp.

Fabi G., Spagnolo A. 1995. Allevamento sperimentale di orate in Medio Adriatico. Gazzettino della Pesca, 11: 4-7 pp.

Fabi G., Spagnolo A. 2001. Le barriere artificiali. Pagg. 454-466 in Cataudella S., Bronzi P. (eds.), Acquacoltura Responsabile. Unimar-Uniprom, Roma.

Falace A., Bressan G. 1990. Dinamica della colonizzazione algale di una barriera artificiale sommersa nel Golfo di Trieste: macrofouling. Hydrores, 7(8): 5-27 pp.

Falace A., Bressan G. 1994. Some observations on periphyton colonization of artificial substrata in the Gulf of Trieste (North Adriatic Sea). Bulletin of Marine Science, 55 (2-3): 924-931 pp.

Falace A., Bressan G. 1995. Adapting an artificial reef to biological requirements. Proceedings ECOSET 1995. Published by Japan International Marine Science and Technology Federation, 2: 634-639 pp.

Falace A., Bressan G. 1995. Esperienze di strutture artificiali sommerse nel Golfo di Trieste (Nord Adriatico). Biologia Marina Mediterranea,  2(1): 123-128 pp.

Falace A., Bressan G. 1997. Adapting an artificial reef to biological requirements.Pages 307-311. In: Hawkins L.E. et al. Proceedings of the 30th European Marine Biology Symposium, 18-20 September.

Falace A., Bressan G. 1999. Quantitative evaluation of algal community on an artificial reef in the Gulf of Trieste (Northern Adriatic Sea). Proceedings Seventh International Conference on Artificial Reefs and related Acquatic Habitats (7th CARAH), Sanremo 1999: 173-178 pp.

Falace A., Zanelli E., Bressan G. 2006. Algal transplantation as a potential tool for artificial reef management and environmental mitigation. Bulletin of Marine Science, 78 (1): 161-166 pp.

Fava F., Ponti M., Giovanardi O., Abbiati M. 2010. Benthic assemblage on concrete artificial reefs in the Northern Adriatic Sea: comparison between Tecnoreef pyramids and cubic bundles of tubes. Rapport de la Commission internationale pour l'exploration scientifique de la Méditerranée, 38: 511 pp.

Franceschini G., Antonini C., Sabatini L., Giovanardi O., 2008. La fauna ittica commerciale della Zona di Tutela Biologica Di Chioggia. Abstract book, Workshop Pesca e Gestione delle Aree Marine Protette, Porto Cesareo, 30 e 31 ottobre 2008.

Franceschini G., Giovanardi O., 2005. La ricerca applicata alla pesca nella zona di tutela biologica (ZTB) di Chioggia. Atti del 1° Convegno Subacquea & Ambiente: le Tegnùe di Chioggia, Centro congressi Sottomarina di Chioggia, 17 – 18 settembre 2005.

Franceschini G., Raicevich S., Giovanardi O., Pranovi F., Manzueto L., 2003. Le "tegnùe" di Chioggia: valutazione dell'impatto della pesca a strascico con metodi acustici e sistemi informatici. Chioggia-rivista di studi e ricerche, 23: 92-102 pp.

Frka D. & Mesić J. Tajne Jadrana.

Frka D. & Mesić J. 2012. Blago Jadrana.

Froglia C., Gramitto M.E. 1998. Osservazioni sull'alimentazione di Sciaena umbra ed Umbrina cirrosa (Pisces, Sciaenidae) in prossimità di barriere artificiali in Adriatico. Biologia Marina Mediterranea, 5 (1): 100-108 pp.

Giansante C., Fatigati M., Ciarrocchi F., Milillo G.S., Onori L., Ferri N. 2010. Monitoring of ichthyic fauna in artificial reefs along the Adriatic coast of the Abruzzi Region of Italy. Veterinaria Italiana 46(3), 365-374 pp.

Giovanardi O., Cristofalo G., Manzueto L., Franceschini G. 2003a. Le "Tegnùe" di Chioggia: nuovi dati e osservazioni sulla base di campionamenti acustici ad alta definizione ("Multibeam" e "Side Scan Sonar"). Chioggia - Rivista di studi e ricerche 23: 103-116 pp.

Giovanardi O., Cristofalo G., Manzueto L., Franceschini G., 2003b. New data on biogenic reefs (Tegnue of Chioggia) in Adriatic. In: Özhan, E. Proceedings of the Sixth International Conference on the Mediterranean Coastal Environment, MEDCOAST 03. Ravenna, Italy. Middle East Technical University, Ankara, Turkey: 1895-1904 pp.

Giovanardi O., Franceschini G., Antonini C., Sabatini L., Boscolo S., Ponti M., Fava F., 2010. Valutazione degli effetti della Zona di Tutela Biologica di Chioggia sui popolamenti demersali e bentonici e sulle possibilità di ripopolamento di specie di interesse commerciale. Relazione finale. VI Piano Triennale della pesca e dell'acquacoltura (cod. A086), 160 pp.

Grati F., Scarcella G., Bolognini L., Fabi G. 2011. Releasing of the European sea bass Dicentrarchus labrax (Linnaeus) in the Adriatic Sea: large-volume versus intensively cultured juveniles. Journal of Experimental Marine Biology and Ecology, 397 (2): 144-152 pp.

La Mesa M., Scarcella G., Grati F., Fabi G. 2010. Age and growth of the black scorpionfish, Scorpena porcus (Pisces, Scorpaenidae) from artificial structures and natural reefs in the Adriatic Sea. Sciencia Marina, 74(4): 677-685 pp.

Le BA esistenti in Friuli Venezia Giulia. Manuale per il monitoraggio delle barriere artificiali sommerse. 2006. INTERREG III A - Transfrontaliero Adriatico. Tavolo blu Adriatico per la gestione sostenibile delle attività di pesca e delle risorse alieutiche dell'Adriatico - ADRIBLU. p: 64-65.

Le BA esistenti in Veneto. Manuale per il monitoraggio delle barriere artificiali sommerse. 2006. INTERREG III A - Transfrontaliero Adriatico. Tavolo blu Adriatico per la gestione sostenibile delle attività di pesca e delle risorse alieutiche dell'Adriatico - ADRIBLU p: 64-103.

Le barriere artificiali ADRI.BLU in Friuli Venezia Giulia. AT 4 -Azioni pilota finalizzate all'incremento della sostenibilità delle attività di pesca. Gestione sostenibile delle attività di pesca e delle risorse alieutiche dell'Adriatico. 2007. Progetto ADRIBLU. INTERREG III A - Alto Adriatico. p: 95-105.

Lorito S., Lucani P., Calabrese L., Perini L. 2012 (aggiornamento 2014). MARE... istruzioni per l'uso - versione 2.2. Dir. Ambiente, Servizio geologico, sismico e dei suoli - Regione Emilia-Romagna. 12 pp.

Lucchetti A. 1998. Studio sull'alimentazione di Sciaena umbra L. In prossimità di una barriera artificiale del medio Adriatico. Università Politecnica delle Marche, Thesis Degree.

Maffei M.,Giulini G., Fabi G., Fiorentini L. 1996. Valutazioni comparative su alcuni parametri di qualità in popolazioni di mitili (Mytilus galloprovincialis Lamk.) provenienti da differenti condizioni di allevamento e da banco naturale. Biologia Marina Mediterranea, 3 (1): 242-245 pp.

Manoukian S. 2011. Impacts of Artificial Reefs on Surrounding Ecosystems. University of South Florida. PhD Thesis. 200 pp.

Manoukian S., Fabi G., Naar D.F. 2011. Multibeam investigation of an Artificial reef settlement in the Adriatic Sea (Italy) 33 years after its deployment. Brazilian Journal of Oceanography, 59 (special issue CARAH):145-153 pp.

Manoukian S., Fabi G., Spagnolo A. 2004. Use of Multibeam Echosounder to detect terrain changes around two artificial reefs (Western Adriatic Sea). Rapport de la Commission internationale pour l'exploration scientifique de la Méditerranée, 37: 52 pp.

Mattassi G., Bettoso N., Rossin P. 2008. Gestione sostenibile delle risorse alieutiche marine e lagunari. Sistema georeferenziato GIS e monitoraggio delle barriere artificiali sommerse. Rapporto 2007. Rapporto per l'Agenzia Regionale per la Protezione dell'Ambiente del Friuli Venezia Giulia. 46 pp."

Molin E., Maggiore F., Zanella M. 2006. Stime di biomassa di Tubularia crocea (Agassiz, 1862) mediante monitoraggio fotografico in un'area a barriere artificiali nel nord Adriatico. Biologia Marina Mediterranea 13 (1): 737-740 pp.

Monitoraggio biologico sulle barriere artificiali installate in prossimità dei comuni di Martinsicuro e Alba Adriatica. 1° anno di monitoraggio - anno 2007. 2007. Istituto Zooprofilattico Sperimentale dell'Abruzzo e del Molise ""G. Caporale"" & Provincia di Teramo. Doc.U.P. Pesca - 2000-2006 Mis. 3.1- Provincia di Teramo - Cod. Progetto 01/BA/04/AB ""PROTEZIONE E SVILUPPO DELLE RISORSE ACQUATICHE"" 44 pp.

Monitoraggio biologico sulle barriere artificiali installate in prossimità dei comuni di Martinsicuro e Alba Adriatica. 2° anno di monitoraggio - anno 2008. 2008. Istituto Zooprofilattico Sperimentale dell'Abruzzo e del Molise ""G. Caporale"" & Provincia di Teramo. Doc.U.P. Pesca - 2000-2006 Mis. 3.1- Provincia di Teramo - Cod. Progetto 01/BA/04/AB ""PROTEZIONE E SVILUPPO DELLE RISORSE ACQUATICHE"" 63 pp.

Monitoraggio biologico sulle barriere artificiali installate in prossimità dei comuni di Martinsicuro e Alba Adriatica. 3° anno di monitoraggio - anno 2009. 2009. Istituto Zooprofilattico Sperimentale dell'Abruzzo e del Molise ""G. Caporale"" & Provincia di Teramo. Doc.U.P. Pesca - 2000-2006 Mis. 3.1- Provincia di Teramo - Cod. Progetto 01/BA/04/AB ""PROTEZIONE E SVILUPPO DELLE RISORSE ACQUATICHE"" 51 pp.

Monitoraggio biologico sulle barriere artificiali installate in prossimità dei comuni di Martinsicuro e Alba Adriatica. 4° anno di monitoraggio - anno 2010. 2010. Istituto Zooprofilattico Sperimentale dell'Abruzzo e del Molise ""G. Caporale"" & Provincia di Teramo. Doc.U.P. Pesca - 2000-2006 Mis. 3.1- Provincia di Teramo - Cod. Progetto 01/BA/04/AB ""PROTEZIONE E SVILUPPO DELLE RISORSE ACQUATICHE"" 62 pp.

Monitoraggio biologico sulle barriere artificiali installate in prossimità dei comuni di Martinsicuro e Alba Adriatica. 5° anno di monitoraggio - anno 2011. 2011. Istituto Zooprofilattico Sperimentale dell'Abruzzo e del Molise ""G. Caporale"" & Provincia di Teramo. Doc.U.P. Pesca - 2000-2006 Mis. 3.1- Provincia di Teramo - Cod. Progetto 01/BA/04/AB ""PROTEZIONE E SVILUPPO DELLE RISORSE ACQUATICHE"" 61 pp.

Monitoraggio biologico sulle barriere artificiali installate in prossimità dei comuni di Martinsicuro e Alba Adriatica. 6° anno di monitoraggio - anno 2012. 2012. Istituto Zooprofilattico Sperimentale dell'Abruzzo e del Molise ""G. Caporale"" & Provincia di Teramo. Doc.U.P. Pesca - 2000-2006 Mis. 3.1- Provincia di Teramo - Cod. Progetto 01/BA/04/AB ""PROTEZIONE E SVILUPPO DELLE RISORSE ACQUATICHE"" 65 pp.

Monitoraggio biologico sulle barriere artificiali installate in prossimità dei comuni di Martinsicuro e Alba Adriatica. 8° anno di monitoraggio - anno 2015. 2015. Istituto Zooprofilattico Sperimentale dell'Abruzzo e del Molise ""G. Caporale"" & Provincia di Teramo. Doc.U.P. Pesca - 2000-2006 Mis. 3.1- Provincia di Teramo - Cod. Progetto 01/BA/04/AB ""PROTEZIONE E SVILUPPO DELLE RISORSE ACQUATICHE"" 64 pp."

Monitoraggio biologico sulle barriere artificiali installate in prossimità della Torre del Cerrano. 2° anno di monitoraggio - anno 2006. 2006. Istituto Zooprofilattico Sperimentale dell'Abruzzo e del Molise ""G. Caporale"" & Provincia di Teramo. Doc.U.P. Pesca - 2000-2006 Mis. 3.1- Provincia di Teramo - Cod. Progetto 04/BA/02/AB. 59 pp.

Monitoraggio biologico sulle barriere artificiali installate in prossimità della Torre del Cerrano. 3° anno di monitoraggio - anno 2007. 2007. Istituto Zooprofilattico Sperimentale dell'Abruzzo e del Molise ""G. Caporale"" & Provincia di Teramo. Doc.U.P. Pesca - 2000-2006 Mis. 3.1- Provincia di Teramo - Cod. Progetto 04/BA/02/AB. 51 pp.

Monitoraggio biologico sulle barriere artificiali installate in prossimità della Torre del Cerrano. 4° anno di monitoraggio - anno 2008. 2008. Istituto Zooprofilattico Sperimentale dell'Abruzzo e del Molise ""G. Caporale"" & Provincia di Teramo. Doc.U.P. Pesca - 2000-2006 Mis. 3.1- Provincia di Teramo - Cod. Progetto 04/BA/02/AB. 69 pp.

Monitoraggio biologico sulle barriere artificiali installate in prossimità della Torre del Cerrano. 5° anno di monitoraggio - anno 2009. 2009. Istituto Zooprofilattico Sperimentale dell'Abruzzo e del Molise ""G. Caporale"" & Provincia di Teramo. Doc.U.P. Pesca - 2000-2006 Mis. 3.1- Provincia di Teramo - Cod. Progetto 04/BA/02/AB. 52 pp.

Monitoraggio biologico sulle barriere artificiali installate in prossimità della Torre del Cerrano. 6° anno di monitoraggio - anno 2011. 2011. Istituto Zooprofilattico Sperimentale dell'Abruzzo e del Molise ""G.

Caporale"" & Provincia di Teramo. Doc.U.P. Pesca - 2000-2006 Mis. 3.1- Provincia di Teramo - Cod. Progetto 04/BA/02/AB. 60 pp.

Monitoraggio biologico sulle barriere artificiali installate in prossimità della Torre del Cerrano. 7° anno di monitoraggio - anno 2012. 2012. Istituto Zooprofilattico Sperimentale dell'Abruzzo e del Molise ""G. Caporale"" & Provincia di Teramo. Doc.U.P. Pesca - 2000-2006 Mis. 3.1- Provincia di Teramo - Cod. Progetto 04/BA/02/AB. 73 pp.

Monitoraggio biologico sulle barriere artificiali installate in prossimità della Torre del Cerrano. 8° anno di monitoraggio - anno 2014. 2014. Istituto Zooprofilattico Sperimentale dell'Abruzzo e del Molise ""G. Caporale"" & Provincia di Teramo. Doc.U.P. Pesca - 2000-2006 Mis. 3.1- Provincia di Teramo - Cod. Progetto 04/BA/02/AB. 63 pp.

Monitoraggio biologico sulle barriere artificiali installate in prossimità della Torre del Cerrano. 9° anno di monitoraggio - anno 2015. 2015. Istituto Zooprofilattico Sperimentale dell'Abruzzo e del Molise ""G. Caporale"" & Provincia di Teramo. Doc.U.P. Pesca - 2000-2006 Mis. 3.1- Provincia di Teramo - Cod. Progetto 04/BA/02/AB. 58 pp.

Monitoraggio biologico sulle barriere artificiali installate in prossimità di Cologna. 2° anno di monitoraggio - anno 2006. 2006. Istituto Zooprofilattico Sperimentale dell'Abruzzo e del Molise ""G. Caporale"" & Provincia di Teramo. Doc.U.P. Pesca - 2000-2006 Mis. 3.1- Provincia di Teramo - Cod. Progetto 03/BA/03/AB. 71 pp.

Monitoraggio biologico sulle barriere artificiali installate in prossimità di Cologna. 3° anno di monitoraggio - anno 2007. 2007. Istituto Zooprofilattico Sperimentale dell'Abruzzo e del Molise ""G. Caporale"" & Provincia di Teramo. Doc.U.P. Pesca - 2000-2006 Mis. 3.1- Provincia di Teramo - Cod. Progetto 03/BA/03/AB. 60 pp.

Monitoraggio biologico sulle barriere artificiali installate in prossimità di Cologna. 4° anno di monitoraggio - anno 2008. 2008. Istituto Zooprofilattico Sperimentale dell'Abruzzo e del Molise ""G. Caporale"" & Provincia di Teramo. Doc.U.P. Pesca - 2000-2006 Mis. 3.1- Provincia di Teramo - Cod. Progetto 03/BA/03/AB. 61 pp.

Monitoraggio biologico sulle barriere artificiali installate in prossimità di Cologna. 5° anno di monitoraggio - anno 2009. 2009. Istituto Zooprofilattico Sperimentale dell'Abruzzo e del Molise ""G. Caporale"" & Provincia di Teramo. Doc.U.P. Pesca - 2000-2006 Mis. 3.1- Provincia di Teramo - Cod. Progetto 03/BA/03/AB. 54 pp.

Monitoraggio biologico sulle barriere artificiali installate in prossimità di Cologna. 6° anno di monitoraggio - anno 2011. 2011. Istituto Zooprofilattico Sperimentale dell'Abruzzo e del Molise ""G. Caporale"" & Provincia di Teramo. Doc.U.P. Pesca - 2000-2006 Mis. 3.1- Provincia di Teramo - Cod. Progetto 03/BA/03/AB. 57 pp.

Monitoraggio biologico sulle barriere artificiali installate in prossimità di Cologna. 7° anno di monitoraggio - anno 2012. 2012. Istituto Zooprofilattico Sperimentale dell'Abruzzo e del Molise ""G. Caporale"" & Provincia di Teramo. Doc.U.P. Pesca - 2000-2006 Mis. 3.1- Provincia di Teramo - Cod. Progetto 03/BA/03/AB. 68 pp.

Monitoraggio biologico sulle barriere artificiali installate in provincia di Pescara. 1° anno di monitoraggio - anno 2005. 2005. Istituto Zooprofilattico Sperimentale dell'Abruzzo e del Molise ""G. Caporale"" & Provincia di Teramo. Doc.U.P. Pesca - 2000-2006 Mis. 3.1- Provincia di Pescara - Cod. Progetto 03/BA/02/AB - ""PROTEZIONE E SVILUPPO DELLE RISORSE ACQUATICHE"". 56 pp.

Monitoraggio biologico sulle barriere artificiali installate in provincia di Pescara. 2° anno di monitoraggio - anno 2006. 2006. Istituto Zooprofilattico Sperimentale dell'Abruzzo e del Molise ""G. Caporale"" & Provincia di Teramo. Doc.U.P. Pesca - 2000-2006 Mis. 3.1- Provincia di Pescara - Cod. Progetto 03/BA/02/AB - ""PROTEZIONE E SVILUPPO DELLE RISORSE ACQUATICHE"". 74 pp.

Monitoraggio biologico sulle barriere artificiali installate in provincia di Pescara. 3° anno di monitoraggio - anno 2007. 2007. Istituto Zooprofilattico Sperimentale dell'Abruzzo e del Molise ""G. Caporale"" & Provincia

di Teramo. Doc.U.P. Pesca - 2000-2006 Mis. 3.1- Provincia di Pescara - Cod. Progetto 03/BA/02/AB - ""PROTEZIONE E SVILUPPO DELLE RISORSE ACQUATICHE"". 68 pp.

Monitoraggio biologico sulle barriere artificiali installate in provincia di Pescara. 4° anno di monitoraggio - anno 2008. 2008. Istituto Zooprofilattico Sperimentale dell'Abruzzo e del Molise ""G. Caporale"" & Provincia di Teramo. Doc.U.P. Pesca - 2000-2006 Mis. 3.1- Provincia di Pescara - Cod. Progetto 03/BA/02/AB - ""PROTEZIONE E SVILUPPO DELLE RISORSE ACQUATICHE"". 100 pp.

Monitoraggio biologico sulle barriere artificiali installate in provincia di Pescara. 5° anno di monitoraggio - anno 2009. 2009. Istituto Zooprofilattico Sperimentale dell'Abruzzo e del Molise ""G. Caporale"" & Provincia di Teramo. Doc.U.P. Pesca - 2000-2006 Mis. 3.1- Provincia di Pescara - Cod. Progetto 03/BA/02/AB - ""PROTEZIONE E SVILUPPO DELLE RISORSE ACQUATICHE"". 66 pp.

Monitoraggio biologico sulle barriere artificiali installate in provincia di Pescara. 6° anno di monitoraggio - anno 2010. 2010. Istituto Zooprofilattico Sperimentale dell'Abruzzo e del Molise ""G. Caporale"" & Provincia di Teramo. Doc.U.P. Pesca - 2000-2006 Mis. 3.1- Provincia di Pescara - Cod. Progetto 03/BA/02/AB - ""PROTEZIONE E SVILUPPO DELLE RISORSE ACQUATICHE"". 66 pp.

Monitoraggio biologico sulle barriere artificiali installate in provincia di Pescara. 7° anno di monitoraggio - anno 2011. 2011. Istituto Zooprofilattico Sperimentale dell'Abruzzo e del Molise ""G. Caporale"" & Provincia di Teramo. Doc.U.P. Pesca - 2000-2006 Mis. 3.1- Provincia di Pescara - Cod. Progetto 03/BA/02/AB - ""PROTEZIONE E SVILUPPO DELLE RISORSE ACQUATICHE"". 66 pp.

Monitoraggio biologico sulle barriere artificiali installate in provincia di Pescara. 8° anno di monitoraggio - anno 2012. 2012. Istituto Zooprofilattico Sperimentale dell'Abruzzo e del Molise ""G. Caporale"" & Provincia di Teramo. Doc.U.P. Pesca - 2000-2006 Mis. 3.1- Provincia di Pescara - Cod. Progetto 03/BA/02/AB - ""PROTEZIONE E SVILUPPO DELLE RISORSE ACQUATICHE"". 92 pp.

Monitoraggio biologico sulle barriere artificiali installate in provincia di Pescara. 9° anno di monitoraggio - anno 2013. 2013. Istituto Zooprofilattico Sperimentale dell'Abruzzo e del Molise ""G. Caporale"" & Provincia di Teramo. Doc.U.P. Pesca - 2000-2006 Mis. 3.1- Provincia di Pescara - Cod. Progetto 03/BA/02/AB - ""PROTEZIONE E SVILUPPO DELLE RISORSE ACQUATICHE"". 67 pp.

Monitoraggio biologico sulle barriere artificiali installate in provincia di Pescara. 10° anno di monitoraggio - anno 2014. 2014. Istituto Zooprofilattico Sperimentale dell'Abruzzo e del Molise ""G. Caporale"" & Provincia di Teramo. Doc.U.P. Pesca - 2000-2006 Mis. 3.1- Provincia di Pescara - Cod. Progetto 03/BA/02/AB - ""PROTEZIONE E SVILUPPO DELLE RISORSE ACQUATICHE"". 96 pp.

Monitoring program associated to the construction of the AR (2005-2014), Benthic community, Fish community, Heavy metals, PAH in mussels, Planktonic communities, Sediments, Water column. Controllo scientifico delle aree sperimentali poste nel mare Adriatico entro le tre miglia lungo la costa teatina in Comune di Vasto - Relazione finale anni 2005-2006. 2006. Agenzia Regionale per la Tutela dell'Ambiente - ARTA Abruzzo. Cod. Progetto 02/BA/02/AB - ""PROTEZIONE E SVILUPPO DELLE RISORSE ACQUATICHE. 74 pp.

Monitoring program associated to the construction of the AR (2006 - 2015), Benthic community, Fish community, Heavy metals, PAH in mussels, Planktonic communities, Sediments, Water column. Progetto per la realizzazione di un'area da destinare allo sviluppo e protezione delle risorse acquatiche in provincia di Chieti prospiciente il Comune di Rocca San Giovanni - Monitoraggio delle risorse eco-biologiche ed alieutiche. 1° anno di monitoraggio - anno 2006. 2006. Agenzia Regionale per la Tutela dell'Ambiente - ARTA Abruzzo. Cod. Progetto 02/BA/04/AB - ""PROTEZIONE E SVILUPPO DELLE RISORSE ACQUATICHE"". 96 pp.

Monitoring program associated to the construction of the AR (2006-2016), Benthic community, Fish community, Water column. Monitoraggio biologico sulle barriere artificiali installate in prossimità della Torre

del Cerrano. 1° anno di monitoraggio - anno 2005. 2005. Istituto Zooprofilattico Sperimentale dell'Abruzzo e del Molise ""G. Caporale"" & Provincia di Teramo. Doc.U.P. Pesca - 2000-2006 Mis. 3.1- Provincia di Teramo - Cod. Progetto 04/BA/02/AB. 49 pp.

Odorico R., 1994. Riserva marina di Miramare: osservazioni subacquee sullo sviluppo da seme di Cymodocea nodosa. Annuario Hydrores, 11: 12–13 pp.

Odorico R., Bussani M., Mora G. 1999. Use of polyethylene in artificial structures: Ecomare Project. Proceedings Seventh International Conference on Artificial Reefs and related Acquatic Habitats (7th CARAH), Sanremo 1999: 652-654 pp.

Orel G. 1988. Aspetti della bionomia bentonica e della pesca del Golfo di Trieste con particolare riferimento ai fondali prospicienti il promontorio di Miramare. Hydrores V(6): 57-70 pp.

Orel G., de Walderstein W., Landri P. Monitoraggio quinquennale della Zona Marina Protetta del Primero. 3° stato avanzamento lavori.

Orel G., de Walderstein W., Zamboni R., Grim F. 2000. Realizzazione della Zona Marina Protetta del Primero mediante l'impiego di Strutture Sommerse MultiLivello. 2° Convegno Nazionale delle Scienze del Mare. Genova, ottobre 2000.

Pacchioli D., Ponti M. 2002. Censimenti subacquei. I progetti in Adriatico settentrionale. Deep, Anno V, maggio-giugno 2002, n.21, 5-7 pp.

Panfili M. 1996. Evoluzione del popolamento ittico in una barriera artificiale del Medio Adriatico. Uiversità degli Studi di Milano, Thesis  Degree. 100 pp.

Piano di monitoraggio ambientale (fase di esercizio) del Terminale GNL di Porto Viro e della condotta di collegamento alla terraferma. ISPRA 2016. Relazione tecnico scientifica Indagini Remotely Operated Vehicle (ROV). Area del Terminale -Fase di esercizio provvisorio V anno (46 E), pp. 53.

Pivetta S., Spazzapan G. 2012. Relitti e navi sommerse. La costa adriatica e ionica dalla Venezia Giulia alla Puglia. Guida ai relitti moderni nei mari italiani. ISBN 978-88-6649-011-1.

Ponti M. 2001. Aspetti biologici ed ecologici delle ""tegnùe"": biocostruzione, biodiversità e salvaguardia. Chioggia - rivista di studi e ricerche 18: 179-194 pp.

Ponti M., Abbiati M., Seccherelli V.U. 2002. Drilling platforms as artificial reefs: distribution of macrobenthic assemblages of the ""Paguro"" wreck (northern Adriatic Sea). ICES Journal of Marine Science 59. S316-S323.

Ponti M., Capra A., Gabbianelli G., Ceccherelli V.U. 1998. Environmental characterization and macrobenthic communities on the Northern Adriatic ""Paguro"" wreck. Rapp. Comm. int. Mer Médit., 35, 1998, 478 pp.

Ponti  M., Fava F., Fabi G., Giovanardi O. 2010. Benthic Assemblages on artificial pyramids along the Central and Northern Adriatic italian coasts. Biologia Marina Mediterranea, 17 (1): 177-178 pp.

Ponti M., Fava F., Perlini R.A., Giovanardi O., Abbiati M. 2015. Benthic assemblages on artificial reefs in the northwestern Adriatic Sea: Does structure type and age matter? Marine Environmental Research, 104: 10-19.

Ponti M., Mastrototaro F. 2005. Distribuzione dei popolamenti ad ascidie sui fondali rocciosi (Tegnùe) al largo di Chioggia (Venezia). 149. In: Riassunti del 36° Congresso nazionale della Società Italiana di Biologia Marina. Trieste. SIBM 324 pp.

Ponti M., Tumedei M., Colosio F., Abbiati M. 2005. Distribuzione dei popolamenti epibentonici sui fondali rocciosi (Tegnùe) al largo di Chioggia (Venezia). 150. In: Riassunti del 36° Congresso nazionale della Società Italiana di Biologia Marina. Trieste. SIBM 324 pp.

Ponti M., Tumedei M., Colosio F., Abbiati M., 2006. Distribuzione dei popolamenti epibentonici sui fondali rocciosi (Tegnùe) al largo di Chioggia (Venezia). Biol. Mar. Medit., 13(1): 625-628 pp.

Prestinenze G., 2009. Struttura e dinamica di popolazione di Pinna nobilis L., 1758 (Mollusca Bivalvia) all'interno dell'Area Marina Protetta di Miramare (Golfo di Trieste). Master Tesi, Univ. Trieste, Trieste, 96 pp.

Progetto BIOMAP - Biocostruzioni marine in Puglia - P.O. FESR 2007/2013 - ASSE IV - LINEA 4.4 - Azione 4.4.1 (http://www.sit.puglia.it/portal/portale_rete_ecologica/biomap).

Progetto Le trezze dell'alto Adriatico: Studio di alcune aree di particolare pregio ambientale ai fini della valorizzazione delle risorse alieutiche locali. Gordini E. 2010. Relazione tecnico-scientifica finale N. 103 / 2010 RIMA 17 GEA. p: 19-39.

Progetto per la realizzazione di un'area da destinare allo sviluppo e protezione delle risorse acquatiche in provincia di Chieti prospiciente i Comuni di Ortona e S. Vito Chietino - Monitoraggio delle risorse eco-biologiche ed alieutiche. Relazione finale - anno 2006. 2006. Agenzia Regionale per la Tutela dell'Ambiente - ARTA Abruzzo. Cod. Progetto 02/BA/03/AB ""PROTEZIONE E SVILUPPO DELLE RISORSE ACQUATICHE "" - Progetto per la realizzazione di un area da destinare allo sviluppo e protezione delle risorse acquatiche nella Provincia di Chieti prospiciente il Comune di Ortona e san Vito Chietino. "" Monitoraggio delle risorse alieutiche"". 99 pp.

Progetto per la realizzazione di un'area da destinare allo sviluppo e protezione delle risorse acquatiche in provincia di Chieti prospiciente i Comuni di Ortona e S. Vito Chietino - Monitoraggio delle risorse eco-biologiche ed alieutiche. 2° anno di monitoraggio - anno 2007. 2007. Agenzia Regionale per la Tutela dell'Ambiente - ARTA Abruzzo. Cod. Progetto 02/BA/03/AB ""PROTEZIONE E SVILUPPO DELLE RISORSE ACQUATICHE "" - Progetto per la realizzazione di un area da destinare allo sviluppo e protezione delle risorse acquatiche nella Provincia di Chieti prospiciente il Comune di Ortona e san Vito Chietino. "" Monitoraggio delle risorse alieutiche"". 52 pp.

Progetto per la realizzazione di un'area da destinare allo sviluppo e protezione delle risorse acquatiche in provincia di Chieti prospiciente i Comuni di Ortona e S. Vito Chietino - Monitoraggio delle risorse eco-biologiche ed alieutiche. 3° anno di monitoraggio - anno 2008. 2008. Agenzia Regionale per la Tutela dell'Ambiente - ARTA Abruzzo. Cod. Progetto 02/BA/03/AB ""PROTEZIONE E SVILUPPO DELLE RISORSE ACQUATICHE "" - Progetto per la realizzazione di un area da destinare allo sviluppo e protezione delle risorse acquatiche nella Provincia di Chieti prospiciente il Comune di Ortona e san Vito Chietino. "" Monitoraggio delle risorse alieutiche"". 44 pp.

Progetto per la realizzazione di un'area da destinare allo sviluppo e protezione delle risorse acquatiche in provincia di Chieti prospiciente i Comuni di Ortona e S. Vito Chietino - Monitoraggio delle risorse eco-biologiche ed alieutiche. 4° anno di monitoraggio - anno 2009. 2009. Agenzia Regionale per la Tutela dell'Ambiente - ARTA Abruzzo. Cod. Progetto 02/BA/03/AB ""PROTEZIONE E SVILUPPO DELLE RISORSE ACQUATICHE "" - Progetto per la realizzazione di un area da destinare allo sviluppo e protezione delle risorse acquatiche nella Provincia di Chieti prospiciente il Comune di Ortona e san Vito Chietino. "" Monitoraggio delle risorse alieutiche"". 23 pp.

Progetto per la realizzazione di un'area da destinare allo sviluppo e protezione delle risorse acquatiche in provincia di Chieti prospiciente i Comuni di Ortona e S. Vito Chietino - Monitoraggio delle risorse eco-biologiche ed alieutiche. 5° anno di monitoraggio - anno 2010. 2010. Agenzia Regionale per la Tutela dell'Ambiente - ARTA Abruzzo. Cod. Progetto 02/BA/03/AB ""PROTEZIONE E SVILUPPO DELLE RISORSE

ACQUATICHE "" - Progetto per la realizzazione di un area da destinare allo sviluppo e protezione delle risorse acquatiche nella Provincia di Chieti prospiciente il Comune di Ortona e san Vito Chietino. "" Monitoraggio delle risorse alieutiche"". 24 pp.

Progetto per la realizzazione di un'area da destinare allo sviluppo e protezione delle risorse acquatiche in provincia di Chieti prospiciente i Comuni di Ortona e S. Vito Chietino - Monitoraggio delle risorse eco-biologiche ed alieutiche. 6° anno di monitoraggio - anno 2011. 2011. Agenzia Regionale per la Tutela dell'Ambiente - ARTA Abruzzo. Cod. Progetto 02/BA/03/AB ""PROTEZIONE E SVILUPPO DELLE RISORSE ACQUATICHE "" - Progetto per la realizzazione di un area da destinare allo sviluppo e protezione delle risorse acquatiche nella Provincia di Chieti prospiciente il Comune di Ortona e san Vito Chietino. "" Monitoraggio delle risorse alieutiche"". 24 pp.

Progetto per la realizzazione di un'area da destinare allo sviluppo e protezione delle risorse acquatiche in provincia di Chieti prospiciente i Comuni di Ortona e S. Vito Chietino - Monitoraggio delle risorse eco-biologiche ed alieutiche. 7° anno di monitoraggio - anno 2012. 2012. Agenzia Regionale per la Tutela dell'Ambiente - ARTA Abruzzo. Cod. Progetto 02/BA/03/AB ""PROTEZIONE E SVILUPPO DELLE RISORSE ACQUATICHE "" - Progetto per la realizzazione di un area da destinare allo sviluppo e protezione delle risorse acquatiche nella Provincia di Chieti prospiciente il Comune di Ortona e san Vito Chietino. "" Monitoraggio delle risorse alieutiche"". 30 pp."

Progetto per la realizzazione di un'area da destinare allo sviluppo e protezione delle risorse acquatiche in provincia di Chieti prospiciente i Comuni di Vasto e Casalbordino Monitoraggio delle risorse eco-biologiche ed alieutiche. 3° anno di monitoraggio - anno 2007. 2007. Agenzia Regionale per la Tutela dell'Ambiente - ARTA Abruzzo. Cod. Progetto 02/BA/02/AB - ""PROTEZIONE E SVILUPPO DELLE RISORSE ACQUATICHE - Controllo scientifico delle aree sperimentali poste nel mare Adriatico entro le tre miglia lungo la costa teatina in Comune di Vasto."". 45 pp.

Progetto per la realizzazione di un'area da destinare allo sviluppo e protezione delle risorse acquatiche in provincia di Chieti prospiciente i Comuni di Vasto e Casalbordino Monitoraggio delle risorse eco-biologiche ed alieutiche. 4° anno di monitoraggio - anno 2008. 2008. Agenzia Regionale per la Tutela dell'Ambiente - ARTA Abruzzo. Cod. Progetto 02/BA/02/AB - ""PROTEZIONE E SVILUPPO DELLE RISORSE ACQUATICHE - - Controllo scientifico delle aree sperimentali poste nel mare Adriatico entro le tre miglia lungo la costa teatina in Comune di Vasto."". 38 pp.

Progetto per la realizzazione di un'area da destinare allo sviluppo e protezione delle risorse acquatiche in provincia di Chieti prospiciente i Comuni di Vasto e Casalbordino Monitoraggio delle risorse eco-biologiche ed alieutiche. 5° anno di monitoraggio - anno 2009. 2009. Agenzia Regionale per la Tutela dell'Ambiente - ARTA Abruzzo. Cod. Progetto 02/BA/02/AB - ""PROTEZIONE E SVILUPPO DELLE RISORSE ACQUATICHE - - Controllo scientifico delle aree sperimentali poste nel mare Adriatico entro le tre miglia lungo la costa teatina in Comune di Vasto."". 20 pp.

Progetto per la realizzazione di un'area da destinare allo sviluppo e protezione delle risorse acquatiche in provincia di Chieti prospiciente i Comuni di Vasto e Casalbordino Monitoraggio delle risorse eco-biologiche ed alieutiche. 6° anno di monitoraggio - anno 2010. 2010. Agenzia Regionale per la Tutela dell'Ambiente - ARTA Abruzzo. Cod. Progetto 02/BA/02/AB - ""PROTEZIONE E SVILUPPO DELLE RISORSE ACQUATICHE - - Controllo scientifico delle aree sperimentali poste nel mare Adriatico entro le tre miglia lungo la costa teatina in Comune di Vasto."". 21 pp.

Progetto per la realizzazione di un'area da destinare allo sviluppo e protezione delle risorse acquatiche in provincia di Chieti prospiciente i Comuni di Vasto e Casalbordino Monitoraggio delle risorse eco-biologiche ed alieutiche. 7° anno di monitoraggio - anno 2011. 2011. Agenzia Regionale per la Tutela dell'Ambiente - ARTA Abruzzo. Cod. Progetto 02/BA/02/AB - ""PROTEZIONE E SVILUPPO DELLE RISORSE ACQUATICHE - -

Controllo scientifico delle aree sperimentali poste nel mare Adriatico entro le tre miglia lungo la costa teatina in Comune di Vasto."". 20 pp.

Progetto per la realizzazione di un'area da destinare allo sviluppo e protezione delle risorse acquatiche in provincia di Chieti prospiciente i Comuni di Vasto e Casalbordino Monitoraggio delle risorse eco-biologiche ed alieutiche. 8° anno di monitoraggio - anno 2012. 2012. Agenzia Regionale per la Tutela dell'Ambiente - ARTA Abruzzo. Cod. Progetto 02/BA/02/AB - ""PROTEZIONE E SVILUPPO DELLE RISORSE ACQUATICHE - - Controllo scientifico delle aree sperimentali poste nel mare Adriatico entro le tre miglia lungo la costa teatina in Comune di Vasto."". 28 pp.

Progetto per la realizzazione di un'area da destinare allo sviluppo e protezione delle risorse acquatiche in provincia di Chieti prospiciente il Comune di Rocca San Giovanni - Monitoraggio delle risorse eco-biologiche ed alieutiche. 2° anno di monitoraggio - anno 2007. 2007. Agenzia Regionale per la Tutela dell'Ambiente - ARTA Abruzzo. Cod. Progetto 02/BA/04/AB - ""PROTEZIONE E SVILUPPO DELLE RISORSE ACQUATICHE"". 69 pp.

Progetto per la realizzazione di un'area da destinare allo sviluppo e protezione delle risorse acquatiche in provincia di Chieti prospiciente il Comune di Rocca San Giovanni - Monitoraggio delle risorse eco-biologiche ed alieutiche. 3° anno di monitoraggio - anno 2008. 2008. Agenzia Regionale per la Tutela dell'Ambiente - ARTA Abruzzo. Cod. Progetto 02/BA/04/AB - ""PROTEZIONE E SVILUPPO DELLE RISORSE ACQUATICHE"". 51 pp.

Progetto per la realizzazione di un'area da destinare allo sviluppo e protezione delle risorse acquatiche in provincia di Chieti prospiciente il Comune di Rocca San Giovanni - Monitoraggio delle risorse eco-biologiche ed alieutiche. 4° anno di monitoraggio - anno 2009. 2009. Agenzia Regionale per la Tutela dell'Ambiente - ARTA Abruzzo. Cod. Progetto 02/BA/04/AB - ""PROTEZIONE E SVILUPPO DELLE RISORSE ACQUATICHE"". 24 pp.

Progetto per la realizzazione di un'area da destinare allo sviluppo e protezione delle risorse acquatiche in provincia di Chieti prospiciente il Comune di Rocca San Giovanni - Monitoraggio delle risorse eco-biologiche ed alieutiche. 5° anno di monitoraggio - anno 2010. 2010. Agenzia Regionale per la Tutela dell'Ambiente - ARTA Abruzzo. Cod. Progetto 02/BA/04/AB - ""PROTEZIONE E SVILUPPO DELLE RISORSE ACQUATICHE"". 29 pp.

Progetto per la realizzazione di un'area da destinare allo sviluppo e protezione delle risorse acquatiche in provincia di Chieti prospiciente il Comune di Rocca San Giovanni - Monitoraggio delle risorse eco-biologiche ed alieutiche. 6° anno di monitoraggio - anno 2011. 2011. Agenzia Regionale per la Tutela dell'Ambiente - ARTA Abruzzo. Cod. Progetto 02/BA/04/AB - ""PROTEZIONE E SVILUPPO DELLE RISORSE ACQUATICHE"". 24 pp.

Progetto per la realizzazione di un'area da destinare allo sviluppo e protezione delle risorse acquatiche in provincia di Chieti prospiciente il Comune di Rocca San Giovanni - Monitoraggio delle risorse eco-biologiche ed alieutiche. 7° anno di monitoraggio - anno 2012. 2012. Agenzia Regionale per la Tutela dell'Ambiente - ARTA Abruzzo. Cod. Progetto 02/BA/04/AB - ""PROTEZIONE E SVILUPPO DELLE RISORSE ACQUATICHE"". 30 pp.

Punzo E. 2011. Sperimentazione di moduli artificiali per l'allevamento di Pholas dactylus (Bivalvia, Pholadidae) in medio Adriatico. Thesis Degree. 118 pp."

Punzo E., Spagnolo A., Fabi G. 2015. Testing of different materials for artificial reefs (western Adriatic Sea). RECIF Conference of artificial reefs: from materials to ecosystem - ESITC Caen (France), 27-29 January 2015: 44-52 pp.

Punzo E., Spagnolo A., Scarcella G., Santelli A., Sarappa A. 2010. Polychaetes assemblage inside an artificial reef in the Northern Adriatic Sea. (Italy). The tenth international polychaetes conference, Lecce (Italy) 20-26 June 2010. Book of abstracts: 81-82 pp.

Rinaldi A., Rambelli F. 2004. Sul relitto della piattaforma ""Paguro"". Guida al riconoscimento della fauna marina. Editrice La Mandragora, Imola. 222 pp.

Sabatini L., Franceschini G., Giovanardi O. 2008. Abitudini alimentari del sarago sparaglione (Diplodus annularis) nella zona di tutela biologica (ZTB) delle tegnùe di Chioggia. Biol. Mar. Medit., 15(1): 352-353 pp.

Sabatini L., Franceschini G., Giovanardi O. 2013. Parte seconda. Caso di studio del Veneto: Le barriere artificiali antistanti la Sacca del Canarin (RO). Le strutture sommerse per il ripopolamento ittico e la pesca (""barriere artificiali""). Quaderni ISPRA -Ricerca Marina 3-2012. p: 53-114.

Sacaccini A. & Piccinetti C. 1969. Il fondale del mare da Falconara a Tortoreto. Programma di ricerca sulle risorse marine e del fondo marino. Consiglio Nazionale delle ricerche. Serie C.

Sala A. 1997. Valutazione della biomassa vagile in una barriera artificiale con tecnica elettroacustica. Università Politecnica delle Marche, Thesis Degree. 106 pp.

Sala A., Fabi G., Manoukian S., Spagnolo A. 2003. Application of multibeam sonar and stationary hydroacoustic system for 4d marine fish biomass monitoring. ICES Symposium on Fish Behaviour in Exploited Ecosystems. Bergen (Norway), 23-26 giugno 2003. Abstract book: 30.

Santelli A., Punzo E., Scarcella G., Strafella P., Spagnolo A., Fabi G. 2012. Decapods' community associated with an artificial reef (Adriatic Sea). 10th Colloquium Crustacea Decapoda Mediterranea. Athens (Greece) 03-07 June 2012. CD-ROM.

Santelli A., Punzo E., Scarcella G., Strafella P., Spagnolo A., Fabi G. 2013. Decapod Crustaceans associated with an artificial reef (Adriatic Sea). Mediterranean Marine Science, 14 (3): 64-75 pp.

Scarcella G. 2011. Età e accrescimento di Scorpaena porcus Linneo, 1758 e Scorpaena notata Rafinesque, 1810 in Adriatico settentrionale. Università Politecnica delle Marche. PhD thesis.

Scarcella G., Bombace G., Fabi G., Grati F. 2004. Aumento controllato dello sforzo di pesca in una barriera artificiale dell'Adriatico settentrionale: effetti sul popolamento ittico. Biologia Marina Mediterranea, 11 (2): 21-32 pp.

Scarcella G., Grati F., Bolognini L., Domenichetti F., Malaspina S., Manoukian S., Polidori P., Spagnolo A., Fabi G. 2015. Time-series analyses of fish abundance from an artificial reef and a reference area in the Adriatic Sea. Journal of Applied Ichthyology, 31(S3): 74-85.

Scarcella G., Grati F., Polidori P., Domenichetti F., Bolognini L., Fabi G. 2011. Comparison of growth rates estimated by otolith reading of Scorpaena porcus and Scorpaena notata caught on artificial and natural reefs of the Northern Adriatic Sea. Brazilian Journal of Oceanography, 59(special issue CARAH): 33-42 pp.

Scarcella G., La Mesa M., Grati F., Polidori P. 2011. Age and growth of the small red scorpionfish, Scorpaena notata Rafinesque, 1810, based on whole and sectioned otolith readings. Environmental Biology of Fishes, 91: 369-378 pp.

SIC-ZPS IT4070026. Relitto della piattaforma Paguro. Piano di Gestione. 2018. 66 pp. https://ambiente.regione.emilia-romagna.it/it/parchi-natura2000/rete-natura-2000/siti/it4070026

Solustri C. 1998. Studio della comunità bentonica insediata su substrati artificiali nel medio Adriatico. Università Politecnica delle Marche, Thesis Degree.

Spagnolo A. 1993. Osservazioni visive e campionamenti di pesca in una barriera artificiale del Medio Adriatico. Università degli Studi di Pisa, Thesis Degree. 100 pp.

Spagnolo A. 2008. Le Barriere artificiali: esperienze e prospettive. Giornate di incontri su problematiche e su aspetti ambientali sanitari e gestionali della pesca in Adriatico. NPPA Interreg - Cards/Phare ""O.A.S.I.S."" COD. 112, Trani, BA, 01-03/04/08. I Quaderni di OASIS, 5. 24 pp.

Spagnolo A., Cuicchi C., Punzo E., Santelli A., Scarcella G., Fabi G. 2014. Patterns of colonization and succession of benthic assemblages in two artificial substrates. Journal of Sea Research, 88: 78-86 pp.

Spagnolo A., Fabi G. 2006. Sistemi piramidali ""Tecnoreef"" per la costruzione di una barriera artificiale: effetti sul popolamento ittico. Regioni e Ambiente, 6 (VII): 44-46 pp.

Spagnolo A., Fabi G., Manoukian S., Panfili M. 2004. Benthic community settled on an Artificial Reef in the Western Adriatic Sea (Italy). Rapport de la Commission internationale pour l'exploration scientifique de la Méditerranée, 37: 552 pp.

Stefanon A., 2001. Cenni sulla geologia e sugli organismi costruttori delle "tegnùe". Chioggia – rivista di studi e ricerche, 18: 171-177 pp.

Talarico L., Welker C., Frisenda P. 2001. Studio della produzione primaria (ecofisiologia e adattamento di macro-alghe alla luce). - Studio della produttività primaria e della produzione secondaria delle strutture artificiali sommerse poste in prossimità del dosso di S. Croce (Golfo di Trieste, Alto Adriatico), Regione Autonoma Friuli-Venezia Giulia, Ed. Università di Trieste, G. Bressan (a cura di): 69-77 pp.

Tassetti N., Malaspina S., Fabi G. 2015. Using a Multibeam Echosounder to monitor an artificial reef. The International Archives of the Photogrammetry, Remote Sensing and Spatial Information Sciences, Volume XL-5/W5: 207-213 pp.

Tumbiolo M.L., Patti B. 1996. Stima della biomassa bentonica in due barriere artificiali situate nel Mar Adriatico. Biologia Marina Mediterranea, 3: 516-519 pp.

Vazzoler M., Ancona S., Berti L., Zogno A.R., Iacovone V., Fassina D., Bon D., Rossi S., Bertaggia R. Pellizzato M. 2004. Area di protezione e studio del Nord Adriatico: campo sperimentale. Incontro scientifico congiunto CONISMA - AIOL d'intesa con la SIBM e la SITE. Terrasini (PA), 15-22 ottobre 2004. Poster.

Žuljević, Ante; Aleksandar, Barić-Sandro; Emil, Lemac; Tina, Dragutin; Hrvoje, Zekanović; Sara, Kaleb; Nikolić, Vedran; Martina, Markov Podvinski, Vladislav Mihelčić i Anita Babačić Ajduk. Ronjenje u najčudesnijem dijelu Sredozemlja - ronilački vodič podmorjem Šibensko–kninske županije, Šibenik: Javna ustanova za upravljanje zaštićenim područjima i drugim zaštićenim prirodnim vrijednostima na području Šibensko–kninske županije, 2011 (prirucnik)

**Websites:**

http://web.tiscali.it/adriasubancona/20801.html

https://ilpiccolo.gelocal.it/trieste/cronaca/2015/03/22/news/i-relitti-in-fondo-al-mare-di-grado-risorsa-per-tutta-la-regione-1.11114738

https://www.discover-pag.com/hr/

https://www.ilfriuliveneziagiulia.it/relitto-del-mercure-la-battaglia-grado-carlo-beltrame-al-salone-degli-incanti-trieste/

Information about plant and animal species: https://www.biologiamarina.org/

WWW.ECOSEA.EU

---

## Author Response (AR1)

**Answers to Reviewer 1 and modifications addressed in the revised version of the manuscript**

Specific comments:

Q1: The methods and materials are overall described in detail in the paper. I only suggest including a clearer definition of what natural/artificial reefs and wrecks are, just at the beginning of the 2.2.1 section (before line 112). The definition of wrecks is actually included at lines 83 and 140, but a very brief introduction of the distinction of the three elements could be integrated.
A1: We inserted the definition of natural reef, artificial reef, and wreck in the introduction.

Q2: Do you investigate also the www.relitti.it database? This database was suggested by stakeholders within the marine tourism sector (in the frame of an H2020 project) and might be compared to these wrecks data.
A2: We consulted the www.relitti.it website and we found that much information (included coordinates) was not directly available on the website but complex inquiries were required to acquire the data. Moreover, the website was offline during a consistent part of the data collection phase.

Q3: How many questionnaires were collected? only by Partners or also a wider network? This information could be included in line 152.
A3: We shared the questionnaires only with project partners. The number of questionnaires collected will be included in the final version.

Q4: Line 183: the "groups" are mentioned here for the first time after line 31 in the abstract, so a short introduction can be included.
A4: Groups' specifications have been added in paragraph 2.2.1.

Technical corrections:

Line 43 and 53: inserted missing citations.

Line 60: the Reefbase website URL has been included, as requested.

Line 63: the citation has been corrected, as suggested.

Line 91: the hyperlink of the PDF was copy-pasting a bad address mixing it with the line numbers, please make sure to copy-paste the correct spelled address.

Line 97: the URL was misspelled, we corrected it.

Line 150: has been rephrased in order to accomplish the reviewer request.

**Answers to Reviewer 2 and modifications addressed in the revised version of the manuscript**

General comments:

We specified that the study regards the portion of the Adriatic Sea that is composed of Italian, Croatian and International waters in the introduction.

Specific comments:

Q1: Section 2.1 describes a literature and data review: it would be valuable to have these elements as a supplemental information to this paper
A1: The bibliography functional to the dataset enrichment has been released contextually with the revised version of the data paper as supplemental material. All the consulted literature is also available for download with the data from SEANOE repository. More specifically the .xlsx version of the database has a tab called "Literature" reporting for each reef the consulted material (in a relation of "one to many").

Q2: In section 2.2.1, it would be useful to explain more clearly that the 4 questions have been used to categorize reefs in 4 groups and refer to table 2
A2: The four main groups of questions have been evidenced in section 2.2.1, as required.

Q3: In table 2, it would seems more clear to have "Applicability" instead of "Applicability restriction" and simply list the type of elements were the information applies, e.g.: AR; NRs, ARs; ARs, wrecks; ...; all (or explicitly NRs, ARs, wrecks)
A3: The field "Applicability" in Table 2 has been changed accordingly to the suggestions of the reviewer.

Q4: In fig. 4, the fact that missing years are not represented is somehow misleading. I think a different representation would help to better communicate the deployment frequency to readers.
A4: Figure 4 has been corrected including periods where no ARs were deployed and no wrecks sunk, as demanded.

Q5: In the section 6 Data availability (and in a few other places in the manuscript) it is mentioned that the database is available from EMODnet, and then referring to the SEANOE repository. From the documentation (https://www.seanoe.org/html/about.htm) I have understood that SEANOE duplicates records from its repository into the EMODnet Data Ingestion portal but it's not clear whether the specific dataset, described here, has been already included in any of the EMODnet portals/catalogues. Could you please clarify this and update the manuscript accordingly?
A5: The database is currently undergoing the appropriate procedure to be published in the EMODnet catalogs, but it is still in Data Ingestion Portal. We corrected as requested in the appropriate sections.

Q6: In the webGIS interface (https://adrireef.github.io/sandbox3/) there seems to be no explicit way to download the filtered elements after a specific search. This could be a useful added functionality for users. In addition to that, it would be good to have an explicit reference in the webGIS interface both to the original dataset (http://doi.org/10.17882/74880) and the data paper (https://essd.copernicus.org/preprints/essd-2020-384/) to allow interested users to check sources, methods and references.

A6: The WebGIS has been conceived only as a viewer from the original purpose. For sure all the suggested modifications would be valuable for the users and we will consider implementing them. The references to data and the data paper have been added to the WebGIS interface.

Technical correction:

Corrections and suggestions indicated in the supplemental PDF have been incorporated in the revised version of the manuscript.

---

## Author Response (AR2)

Dear Sirs and Madams,

as already stated in previous emails, Figure 6 of this manuscript has been entirely created by me and no map provider has been used.

Since this seems the only remark to our work, I will proceed with the upload of the manuscript in its last shape.

Kind Regards,

Annalisa Minelli